# Embodied Cognition Augmented End2End Autonomous Driving

**Ling Niu**
Tsinghua University
nl23@mails.tsinghua.edu.cn

**Xiaoji Zheng**
Tsinghua University
zhengxj24@mails.tsinghua.edu.cn

**Han Wang**
Tsinghua University
kelsulhgod@gmail.com

**Chen Zheng**
Tsinghua University
zhengchen@air.tsinghua.edu.cn

**Ziyuan Yang**
Tsinghua University
ziyuan86@uw.edu

**Bokui Chen**[*]
Tsinghua University
chenbokui@tsinghua.edu.cn

**Jiangtao Gong**[*]
Tsinghua University
gongjiangtao2@gmail.com

## Abstract

In recent years, vision-based end-to-end autonomous driving has emerged as a new paradigm. However, popular end-to-end approaches typically rely on visual feature extraction networks trained under label supervision. This limited supervision framework restricts the generality and applicability of driving models. In this paper, we propose a novel paradigm termed $E^3AD$, which advocates for comparative learning between visual feature extraction networks and the general EEG large model, in order to learn latent human driving cognition for enhancing end-to-end planning. In this work, we collected a cognitive dataset for the mentioned contrastive learning process. Subsequently, we investigated the methods and potential mechanisms for enhancing end-to-end planning with human driving cognition, using popular driving models as baselines on publicly available autonomous driving datasets. Both open-loop and closed-loop tests are conducted for a comprehensive evaluation of planning performance. Experimental results demonstrate that the $E^3AD$ paradigm significantly enhances the end-to-end planning performance of baseline models. Ablation studies further validate the contribution of driving cognition and the effectiveness of comparative learning process. To the best of our knowledge, this is the first work to integrate human driving cognition for improving end-to-end autonomous driving planning. It represents an initial attempt to incorporate embodied cognitive data into end-to-end autonomous driving, providing valuable insights for future brain-inspired autonomous driving systems. Our code will be made available at https://github.com/AIR-DISCOVER/E-cubed-AD.

---

[*]Corresponding author

39th Conference on Neural Information Processing Systems (NeurIPS 2025).

# 1 INTRODUCTION

Autonomous driving technology is crucial for transferring driving authority from human drivers to sensors and artificial intelligence, promising to enhance efficiency and safety in transportation. In the pursuit of autonomous driving, the emergence of end-to-end autonomous driving has recently garnered increasing attention [10, 12, 26, 31, 16, 14]. However, existing end-to-end autonomous (E2E-AD) driving model frameworks often rely on sequential 3D visual representations, such as Bird's Eye View (BEV) features[18, 17]. BEV features contain rich latent information essential for autonomous driving. However, in existing end-to-end approaches, BEV feature extraction networks are typically supervised using annotated data for downstream tasks such as 3D perception, motion prediction, and semantic segmentation[15, 29]. This limited supervision framework restricts the model's ability to extract visual information.

Models trained only on labeled data are limited, while the human brain uses embodied reasoning to anticipate hazards and adapt to new situations. Recently developed general-purpose EEG models (e.g., LaBraM [13]) can extract rich cognitive features directly from EEG signals and achieve excellent performance across various general tasks. By leveraging the cognitive features provided by such universal brain-inspired models, it becomes possible to offer broader supervision to the feature extraction networks of E2E-AD models.

In this study, we first trained the "Driving-Thinking Model," a spatio-temporal feature extraction network. Unlike traditional supervised approaches relying on manually annotated labels, our model was trained on a specifically collected dataset, utilizing paired video data and corresponding EEG segments. The Driving-Thinking Model is refined through contrastive learning with a large EEG model, enabling it to infer driving-related cognitive information from visual inputs. Subsequently, we explored multiple frameworks to investigate the potential of the driving cognition in enhancing mainstream end-to-end autonomous driving models.

The $E^3AD$ paradigm is the first to incorporate human driving cognition to enhance end-to-end autonomous driving models, yielding significant findings. $E^3AD$ can be directly applied to baseline end-to-end driving models, achieving substantial improvements in driving performance with only a tiny increase in computational cost, while reaching the level of state-of-the-art methods. Moreover, our study investigates the methods and underlying mechanisms by which driving cognition enhances end-to-end planning, making novel contributions to the field of embodied human intelligence augmentation in AI algorithms. At the same time, Our work represents an exploration of a more end-to-end styled autonomous driving framework, enabling the model to acquire richer semantic information from raw data through implicit supervision, rather than being limited to manually annotated labels.

# 2 Related Work

## 2.1 End-to-End Planning

Learning-based planning, particularly reinforcement learning and imitation learning [3, 4, 27], has emerged as a promising approach since Pomerleau's pioneering work [22]. The latest trend in this field is the end-to-end training of multiple functional modules [8, 2]. ST-P3 introduces improvements in perception, prediction, and planning modules, integrating auxiliary tasks such as depth estimation and BEV segmentation to enhance spatio-temporal features learning. UniAD [10] integrated six subtasks (object detection, tracking, map segmentation, trajectory prediction, occupancy prediction, and planning) into a unified network. VAD [12] uses fully vectorized representations of driving scenarios. GenAD formulates autonomous driving as a generative modeling problem, predicting the evolution of the ego vehicle and its surrounding environment based on past scenes, and employs motion and planning heads consistent with VAD. LAW enhances planning by extracting richer spatiotemporal features through a self-supervised paradigm, predicting future scene features based on current features and the ego trajectory. In contrast, our model advocates for contrastive learning between the spatio-temporal feature extraction network and the general EEG model, aiming to acquire potential human driving cognition from visual inputs to further enhance planning.

## 2.2 Cognitive-Enhanced AI Algorithms

Recent research has explored integrating human cognitive data to enhance AI models. In NLP, BrainBERT and CogBERT incorporate brain signals and eye-tracking data, respectively, improving performance [24, 5]. In autonomous driving, world models mimicking human cognitive structures show promise [7], with models like HLTP enhancing trajectory prediction [19]. Liao et al. proposed a model integrating cognitive insights for perceived safety and dynamic decision-making [20]. However, emulating complex human driving processes requires high-quality cognitive data, such as EEG, which provides rich information about intricate cognitive processes. Currently, there is a lack of research exploring whether such high-quality cognitive data can effectively enhance complex tasks like autonomous driving. Recently, the LaBraM framework (Large Brain Model) [13] learns universal EEG representations by self-supervised pre-training on 2,500+ hours of diverse EEG data: it splits signals into channel patches, trains a vector-quantized tokenizer [28], and uses masked-modeling to predict masked tokens. LaBraM sets new state-of-the-art on downstream BCI tasks like anomaly detection and emotion recognition [13]. Due to the good performance of LaBraM, we use it in our work to extract rich and generalizable EEG features which will provide human cognition insights for end2end autonomous driving models.

## 3 METHOD

Given the EEG data are not included in traditional end-to-end autonomous driving training datasets [25, 6, 1], a gap exists between the self-collected dataset and typical autonomous driving datasets. To address this challenge, we propose a two-stage training approach as shown in Fig. 1: EEG data are utilized only during the first stage of training, while in the second stage and during inference, only mainstream autonomous driving datasets are employed to ensure fairness.

1. **Driving-Thinking Model Training**: A visual feature extraction network is trained on a self-collected dataset and supervised using cognitive features from LaBraM [13]. Specifically, inspired by the successful paradigm of CLIP [23], contrastive learning is employed to achieve cross-modal self-supervised learning.

2. **Embodied Cognition Augmented End2End Model Training**: We freeze the Driving-Thinking model and integrate it into popular E2E-AD frameworks. The integrated model is then trained on large-scale driving datasets, followed by open-loop testing and closed-loop simulation. To ensure fairness, only the original data is used at this stage, without introducing any additional inputs, maintaining consistency with other baseline models.

In this section, we will first describe how we train the Driving-Thinking model, followed by the process of integrating it into an end-to-end autonomous driving algorithm.

### 3.1 Embodied Cognitive Dataset

To our knowledge, few cognitive-related datasets have been collected in real-world driving scenarios. Thus, we gathered a multi-modal physiological dataset from 27 subjects driving a fixed route in complex traffic. We recorded CAN bus data, EEG, heart rate, skin conductance, and front-facing camera footage. After filtering data and controlling for variables, we selected 20 male drivers (10 experts, 10 novices).

All EEG recordings were preprocessed in EEGLAB[2] by first re-referencing signals to the M1 and M2 electrodes. We then applied a 0.1–50 Hz band-pass filter to remove drift and high-frequency noise, followed by a 50 Hz notch filter to suppress power-line interference. Next, we employed Independent Component Analysis (ICA) to remove ocular, cardiac, channel, and muscle artifacts, effectively filtering out various noise sources. To match the input requirements of the Large Brain Model[13], the data were downsampled from 1000 Hz to 200 Hz, and amplitudes (±0.1 mV) were normalized by dividing by 0.1 mV, mapping signals into [–1,1]. Finally, each driver's continuous session was manually split into 14 condition-specific segments and uniformly cut into 2 s clips (including those shorter than 2 s as individual samples).

---

[2]https://eeglab.org/

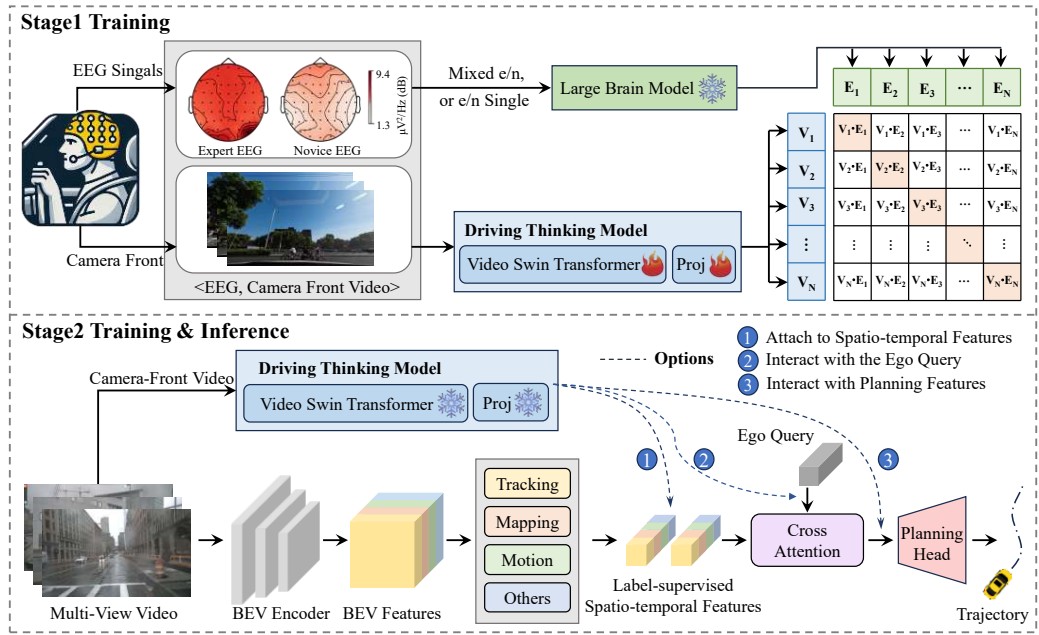

Figure 1: The training is divided into two stages. In the first stage, contrastive learning is conducted between the Driving-Thinking Model—a pretrained spatio-temporal feature extraction network—and the large EEG model LaBraM[13] on a self-collected dataset. In the second stage training and inference processes, we use the same inputs as other driving models, without introducing EEG data. Instead, the entire Driving-Thinking Model is kept frozen. Subsequently, we design three different frameworks to investigate how the driving cognition learned by the Driving-Thinking Model enhances end-to-end planning, as well as the associated mechanisms.

We split the dataset into training set validation, and test set. The structured data was shuffled and divided into two parts with a ratio of 80:10:10. The training set comprised 1894 clips (1037 from expert drivers and 857 from novice drivers), the validation set consists of 236 clips (144 expert, and 92 novice), and the test set contained 237 clips (118 from expert drivers and 119 from novice drivers). For details regarding cognitive data collection and our data, comprehensive information can be found in the Appendix.

### 3.2 Driving–Thinking Model

Let $\{(v_i, e_i)\}_{i=1}^N$ be a minibatch of paired video–EEG samples. We define

$$g_v : \text{Video} \to \mathbb{R}^{d'}, \quad h_v : \mathbb{R}^{d'} \to \mathbb{R}^d, \quad f_e : \text{EEG} \to \mathbb{R}^d,$$

where $g_v$ is implemented as a Video Swin Transformer[21], $h_v$ is a learnable linear projection head, and $f_e$ is the frozen Large Brain Model (LaBraM)[13]. During training only the video-branch parameters $\theta_v = \{g_v, h_v\}$ are updated, while $f_e$ remains fixed.

We begin with feature extraction and normalization: video clips are processed by $g_v$ and $h_v$, and EEG segments by $f_e$, then each is $\ell_2$-normalized:

$$z_i^v = \frac{h_v\big(g_v(v_i)\big)}{\|h_v(g_v(v_i))\|}, \quad z_i^e = \frac{f_e(e_i)}{\|f_e(e_i)\|}. \tag{1}$$

Next, we compute a similarity matrix using a learnable log-temperature $\beta$ (so that $\tau = e^\beta$):

$$s_{ij} = e^\beta \langle z_i^v, z_j^e \rangle, \quad S = [\, s_{ij} \,]_{i,j=1}^N. \tag{2}$$

The model is trained with a symmetric InfoNCE loss:

$$L_v = -\frac{1}{N} \sum_{i=1}^{N} \log \frac{e^{s_{ii}}}{\sum_{j=1}^{N} e^{s_{ij}}},$$

$$L_e = -\frac{1}{N} \sum_{i=1}^{N} \log \frac{e^{s_{ii}}}{\sum_{j=1}^{N} e^{s_{ji}}}, \quad (3)$$

$$L = \tfrac{1}{2}\left(L_v + L_e\right).$$

Finally, we update only the video-branch parameters $\theta_v$ by gradient descent, keeping $f_e$ frozen:

$$\theta_v \leftarrow \theta_v - \eta \, \nabla_{\theta_v} L. \quad (4)$$

### 3.3 Embodied Cognition Augmented End2End Model Training

Perception is crucial in autonomous driving for extracting visual features from sensor data. Among methods, BEV has become popular because it effectively integrates spatial and temporal visual information. As illustrated in Fig. 1, popular end-to-end frameworks inherit the BEV encoder architecture to produce label-supervised spatio-temporal features. These features are further interacted with ego-vehicle queries that encapsulate the historical motion of the ego vehicle, which are then processed by a planning head to generate the final predicted trajectory. It is worth noting that different end-to-end methods exhibit subtle differences in their choices of supervised spatio-temporal features. For example, some approaches[18, 10] adopt the entire BEV feature map as the visual representation. In contrast, other methods[14, 12] propose extracting a sparse features from the BEV feature map, with the aim of enhancing computational efficiency. Despite these divergent design strategies, the overarching framework adopted by mainstream end-to-end methods remains consistent.

Therefore, this study focuses on how human cognition can be integrated to enhance end-to-end planning, as well as the underlying mechanisms. We designed the following three frameworks and corresponding hypotheses:

1. **Attach to Spatio-temporal features**: We hypothesize that the Driving-Thinking model has learned human drivers' attention mechanisms toward visual features. Leveraging this driving cognition, the model can select potential visual features related to planning from raw visual inputs, thereby enhancing end-to-end planning.

2. **Interact with the Ego Query**: We hypothesize that the Driving-Thinking model has acquired driving-related cognitive knowledge about visual features from LaBraM. This knowledge, when interactively embedded into the ego query, can guide the process by which the ego query interacts with visual features to generate planning features.

3. **Interact with planing features**: We hypothesize that the Driving-Thinking model learns more advanced driving cognition directly related to planning tasks. We believe that such higher-level driving cognition can directly interact with the features, improving the preliminary planning results. Therefore, we directly enable its interaction with the features, which is then fed into the planning head for trajectory decoding.

### 3.4 Attach to Spatio-temporal features

BEV features contain visual information aggregated across both spatial and temporal dimensions. Therefore, by interacting the cognitive feature $\mathbf{V}_t$ generated by the Driving-Thinking model with $\mathbf{B}_t$, we can highlight the potential brain-aware BEV feature $\mathbf{B}_t^{brain}$. Considering that BEV features typically have large spatial dimensions, we employ an Attention Gate to generate an attention map based on the max-pooled features of $\mathbf{B}_t$ and $\mathbf{V}_t$, thereby selecting $\mathbf{B}_t^{brain}$.

$$\mathbf{B}_t^{brain} = \mathbf{B}_t \odot AttnGate(Maxpooling(\mathbf{B}_t), \mathbf{V}_t) \quad (5)$$

Both $\mathbf{B}_t^{brain}$ and $\mathbf{B}_t$ share the same spatial dimensions of $b \times hw \times c$, which is not computationally efficient. Therefore, inspired by the success of the BEV TokenLearner, we adopt adaptive spatial attention to obtain brain-aware sparse visual representations $\mathbf{S}_t$. The process is as follows:

$$\mathbf{S}_t = \rho(\varpi(\mathbf{B}_t^{brain})\mathbf{B}_t^{brain})$$
$$\mathbf{S}_t = SelfAttn(\mathbf{S}_t) \quad (6)$$

where the spatial attention function $\varpi$ maps $\mathbf{B}_t^{brain}$ to an attention map for each sparse visual query: $\mathbb{R}^{b \times hw \times c} \to \mathbb{R}^{b \times n_s \times hw}$, where $n_S$ denotes the number of sparse visual queries. The average pooling function $\rho$ is utilized to aggregate spatial features across the $h \times w$ dimensions. Finally, a self-attention layer is applied to enhance the representation capability of $\mathbf{S}_t$. Subsequently, $\mathbf{S}_t$ interacts with the ego-query $\mathbf{Q}_{ego}$ via a decoder-only transformer to generate brain-aware planning features. These features are concatenated with the planning features produced by the end-to-end baseline model, and the combined featuress are then fed into the planning head for trajectory decoding. For HD-map-free planning, a high-level driving command $c$ has been demonstrated to be necessary[12, 10]. Therefore, we adopt the planning head from mainstream end-to-end approaches, and feed the updated features together with the high-level driving command as input to output the planned trajectory $\hat{\mathbf{T}}_t$. The decoding process is formulated as follows:

$$\begin{aligned}
\mathbf{F}_{plan}^b &= CrossAttn(\mathbf{Q}_{ego}, \mathbf{S}_t, \mathbf{S}_t) \\
\mathbf{F}_{plan}^s &= CrossAttn(\mathbf{Q}_{ego}, \mathbf{V}_t^s, \mathbf{V}_t^s) \\
\mathbf{F}_{plan} &= [\mathbf{F}_{plan}^s, \mathbf{F}_{plan}^b] \\
\hat{\mathbf{T}} &= PlanHead(\mathbf{F}_{plan}, cmd = c)
\end{aligned} \tag{7}$$

where, $\mathbf{V}_t^s$ denotes the visual features learned by the end-to-end baseline model through supervised learning with labels. $\mathbf{F}_{plan}^s$ and $\mathbf{F}_{plan}^b$ represent the planning features obtained from $\mathbf{V}_t^s$ and $\mathbf{S}_t$, respectively. The symbol $[\cdot]$ denotes the concatenation operation.

### 3.5 Interaction with the Ego Query

As previously discussed, $\mathbf{Q}_{ego}$ contains only the historical trajectory information of the ego vehicle. According to the second framework and hypothesis described in Section 3.3, the Driving-Thinking model learns the driver's cognition of visual features from the general EEG large model. Therefore, we construct query-key pairs based on $\mathbf{Q}_{ego}$ and the output $\mathbf{V}_t$ of the Driving-Thinking model, and obtain new ego-vehicle query $\mathbf{Q}_{ego}'$ embedding with driving visual cognition through cross-attention. The enriched $\mathbf{Q}_{ego}'$ then interacts with the spatio-temporal features $\mathbf{V}_t^s$, allowing the embedded driving visual cognition to inform the generation of planning feature based on visual information. Similarly, we use the planning head to decode the planned trajectory:

$$\begin{aligned}
\mathbf{Q}_{ego}' &= CrossAttn(\mathbf{Q}_{ego}, \mathbf{V}_t, \mathbf{V}_t) \\
\mathbf{F}_{plan} &= CrossAttn(\mathbf{Q}_{ego}', \mathbf{V}_t^s, \mathbf{V}_t^s) \\
\hat{\mathbf{T}} &= PlanHead(\mathbf{F}_{plan}, cmd = c)
\end{aligned} \tag{8}$$

### 3.6 Interaction with planing features

Recent studies[13] have shown that EEG signals often exhibit changes in specific frequency bands during tasks involving decision-making, attention, and memory. Inspired by these findings, in third framework, we hypothesize that the Driving-Thinking model is capable of acquiring advanced cognitive abilities related to driving decisions and reasoning. This includes human-like reasoning and judgment for preliminary decisions, allowing the model to identify and correct mistakes already in the initial planning stage, and thereby directly influence the quality of the planning feature. To this end, we adopt a multi-layer decoder architecture, which integrates driving-related cognitive information to perform chained reasoning over the initial planning feature $\mathbf{F}_{plan}$. Specifically, $\mathbf{F}_{plan}$ contain information about the trajectory sequence, while $\mathbf{V}_t^s$ is obtained from a spatio-temporal features extraction network. Therefore, we introduce learnable positional encodings to align the latent temporal dependencies present in both $\mathbf{F}_{plan}$ and $\mathbf{V}_t^s$, ultimately producing the planning feature ($\mathbf{F}_{plan}''$) inferred through reasoning combined with human driving cognition.

$$\begin{aligned}
\mathbf{F}_{plan}'' &= TransformerDecoder(q, k, v, q_{pos}, k_{pos}) \\
q &= \mathbf{F}_{plan}, k = v = \mathbf{V}_t \\
\hat{\mathbf{T}} &= PlanHead(\mathbf{F}_{plan}'', cmd = c)
\end{aligned} \tag{9}$$

where, $q_{pos}$ and $k_{pos}$ are learnable positional encodings.

# 4 EXPERIMENTS

## 4.1 Implementation Details

In our implementation, we employ the Video Swin Transformer [21] to extract spatio-temporal visual features from 2s video clips at 2 fps, and the Large Brain Model [13] to extract generic EEG embeddings. Both branches are initialized with pretrained weights, then fine-tuned via a CLIP-style contrastive objective [23]. Each branch's output passes through a two-layer MLP adapter head projecting features into a shared 200-dimensional space. We train the model on aligned 2s video–EEG segment pairs with a batch size of 16 for 120 epochs using Adam (learning rate 2e-5 on both backbones and adapters, lr ratio 1:1), dropout of 0.01, and weight decay of 1e-5. Training on the full dataset requires approximately 12 h on a single NVIDIA A100 40 GB GPU.

Table 1: Open-loop planning performance comparison of different driving models. The ego status was not used in the planning module. † indicates the FPS measured on our NVIDIA A100 GPU, while others use the reported FPS. * denote the best performing models.

| Method | ST-P3 Metrics | | | | | | | | UniAD Metrics | | | | | | | | FPS |
|---|---|---|---|---|---|---|---|---|---|---|---|---|---|---|---|---|---|
| | L2(m) ↓ | | | | Collision(%) ↓ | | | | L2(m) ↓ | | | | Collision(%) ↓ | | | | |
| | 1s | 2s | 3s | Avg. | 1s | 2s | 3s | Avg. | 1s | 2s | 3s | Avg. | 1s | 2s | 3s | Avg. | |
| ST-P3[9] | 1.33 | 2.11 | 2.90 | 2.11 | 0.23 | 0.62 | 1.27 | 0.71 | - | - | - | - | - | - | - | - | 1.6 |
| VAD-Base[12] | 0.41 | 0.70 | 1.05 | 0.72 | 0.07 | 0.17 | 0.41 | 0.22 | - | - | - | - | - | - | - | - | 4.5 |
| VAD-Tiny[12] | 0.46 | 0.76 | 1.12 | 0.78 | 0.21 | 0.35 | 0.58 | 0.38 | - | - | - | - | - | - | - | - | 16.8 |
| UniAD[10] | - | - | - | - | - | - | - | - | 0.48 | 0.96 | 1.65 | 1.03 | 0.05 | 0.17 | 0.71 | 0.31 | 1.8 |
| GenAD[31] | 0.28 | 0.49 | 0.78 | 0.52 | 0.08 | 0.14 | 0.34 | 0.19 | 0.36 | 0.83 | 1.55 | 0.91 | 0.06 | 0.23 | 1.00 | 0.43 | 6.7 |
| BEV-Planner[18] | 0.28 | 0.42 | 0.68 | 0.46 | 0.04 | 0.37 | 1.07 | 0.49 | - | - | - | - | - | - | - | - | _ |
| LAW[16] | 0.26 | 0.57 | 1.01 | 0.61 | 0.14 | 0.21 | 0.54 | 0.30 | - | - | - | - | - | - | - | - | 19.5 |
| VAD-Tiny[12] | 0.46 | 0.76 | 1.12 | 0.78 | 0.21 | 0.35 | 0.58 | 0.38 | - | - | - | - | - | - | - | - | 9.8† |
| $E^3AD$(VAD-Tiny) | 0.40 | 0.66 | 0.98 | 0.68 | 0.18 | 0.33 | 0.55 | 0.35 | - | - | - | - | - | - | - | - | 8.8† |
| VAD-Base[12] | 0.41 | 0.70 | 1.05 | 0.72 | 0.07 | 0.17 | 0.41 | 0.22 | - | - | - | - | - | - | - | - | 3.8† |
| $E^3AD$(VAD-Base) | 0.35 | 0.62 | 0.96 | 0.64 | 0.06 | 0.13 | 0.36 | 0.18* | - | - | - | - | - | - | - | - | 3.7† |
| UniAD[10] | - | - | - | - | - | - | - | - | 0.48 | 0.96 | 1.65 | 1.03 | 0.05 | 0.17 | 0.71 | 0.31 | 1.4† |
| $E^3AD$(UniAD) | - | - | - | - | - | - | - | - | 0.48 | 0.96 | 1.64 | 1.03 | 0.07 | 0.10 | 0.52 | 0.23* | 1.4† |
| GenAD(official checkpoint)[31] | 0.25 | 0.46 | 0.76 | 0.49 | 0.11 | 0.21 | 0.45 | 0.26 | 0.33 | 0.81 | 1.58 | 0.91 | 0.06 | 0.43 | 1.19 | 0.56 | 6.7† |
| GenAD(Reproduce) | 0.25 | 0.46 | 0.76 | 0.49 | 0.14 | 0.26 | 0.50 | 0.30 | 0.33 | 0.79 | 1.58 | 0.90 | 0.17 | 0.49 | 1.15 | 0.60 | 6.7† |
| $E^3AD$(GenAD) | 0.24 | 0.44 | 0.74 | 0.47 | 0.10 | 0.21 | 0.42 | 0.24 | 0.32 | 0.78 | 1.52 | 0.87 | 0.15 | 0.35 | 1.09 | 0.53 | 6.6† |
| LAW[16] | 0.26 | 0.57 | 1.01 | 0.61 | 0.14 | 0.21 | 0.54 | 0.30 | - | - | - | - | - | - | - | - | 16.5† |
| $E^3AD$(LAW) | 0.28 | 0.57 | 0.98 | 0.61 | 0.11 | 0.13 | 0.42 | 0.22 | - | - | - | - | - | - | - | - | 16.1† |

Table 2: Open-loop and Closed-loop Results of E2E-AD Methods in Bench2Drive under base training set.

| Method | Open-loop Metric | Closed-loop Metric | |
|---|---|---|---|
| | Avg.L2(m) ↓ | DS ↑ | SR(%) ↑ |
| AD-MLP | 3.64 | 18.05 | 0.00 |
| UniAD-Tiny | 0.80 | 40.73 | 13.18 |
| UniAD-Base | 0.73 | 45.81 | 16.36 |
| VAD | 0.91 | 42.35 | 15.00 |
| $E^3AD$(UniAD-Base) | **0.69** | **50.07** | **20.12** |
| $E^3AD$(VAD) | **0.86** | **47.63** | **19.54** |

Table 3: Ablation study on the impact of the Driving-Thinking model on driving models is conducted, using the $E^3AD(VAD-Base)$ as the experimental baseline.

| Contrastive Learning | | Freeze | L2(m) ↓ | | | | Collision(%) ↓ | | | |
|---|---|---|---|---|---|---|---|---|---|---|
| Expert | Novice | Video Encoder | 1s | 2s | 3s | Avg. | 1s | 2s | 3s | Avg. |
| - | - | ✓ | 0.39 | 0.68 | 1.02 | 0.70 | 0.13 | 0.21 | 0.40 | 0.25 |
| ✓ | ✓ | - | 0.40 | 0.64 | 1.04 | 0.69 | 0.15 | 0.18 | 0.42 | 0.25 |
| ✓ | - | ✓ | 0.33 | 0.59 | 0.92 | **0.61** | 0.05 | 0.18 | 0.40 | **0.21** |
| - | ✓ | ✓ | 0.38 | 0.65 | 0.99 | 0.67 | 0.07 | 0.18 | 0.38 | 0.21 |
| ✓ | ✓ | ✓ | 0.35 | 0.62 | 0.96 | **0.64** | 0.06 | 0.13 | 0.36 | **0.18** |

Table 4: Comparison of different Frameworks, using the $E^3AD(VAD-Base)$ as the baseline.

| Framework | L2(m) $\downarrow$ | | | | Collision(%)$\downarrow$ | | | |
|---|---|---|---|---|---|---|---|---|
| | 1s | 2s | 3s | Avg. | 1s | 2s | 3s | Avg. |
| Attach to Spatio-temporal features | 0.38 | 0.67 | 1.02 | 0.69 | 0.09 | 0.17 | 0.39 | 0.22 |
| Interact with the Ego Query | 0.37 | 0.66 | 1.04 | 0.69 | 0.06 | 0.14 | 0.41 | 0.20 |
| Interact with the Planning Features | 0.35 | 0.62 | 0.96 | 0.64 | 0.06 | 0.13 | 0.36 | 0.18 |

## 4.2 Dataset and Metrics

We conducted experiments on the publicly available nuScenes dataset [1], which provides 1,000 urban driving scenes under diverse weather conditions. The dataset includes 700 training, 150 validation, and 150 test scenes, each lasting about 20 s with keyframes annotated at 2 Hz. It offers multi-sensor inputs, including six cameras with 360° horizontal FOV, a LiDAR, radar, and IMUs. Following established end-to-end model testing methodologies [9, 10, 12], we employ $l_2$ displacement errors and collision rates as metrics to evaluate the quality of the planning process. For closed-loop evaluation, we use Bench2Drive[11], which under CARLA Leaderboard 2.0 for end-to-end autonomous driving. It provides an official training set, where we use the base set (1000 clips) for fair comparison with all the other baselines. We use the official 220 routes for evaluation.

## 4.3 Main Result

**Open-Loop Evaluation** Our method achieves outstanding performance on the NuScenes dataset, as shown in Tab. 1. In line with recent studies, we recognize that the L2 error primarily reflects model convergence, yet we report it for completeness. According to the experimental results, after applying the $E^3AD$ method, the average collision rates of UniAD and VAD-Base decreased by 0.08% (25.8% relatively) and 0.04% (18.2% relatively), respectively, even surpassing the recent state-of-the-art methods. In addition, the convergence L2 errors were reduced by 0.08 (11.1% relatively) and 0.05 m (6.7% relatively), respectively. Moreover, the $E^3AD$ method is also applicable to the lightweight VAD-Tiny model, where the average collision rate and the L2 error was significantly discreased. It is worth noting that, beyond the aforementioned autoregressive trajectory generation framework, our approach generalizes well to other paradigms, including GenAD, which employs a VAE-based trajectory generation mechanism, and LAW, which leverages world model theory for self-supervised BEV feature learning. In both cases, our method yields consistent improvements compared to the official checkpoints and our reproduced results. Despite these improvements, our method maintains the efficient inference speed of lightweight driving models.

**Closed-Loop Evaluation** As presented in Tab. 2. In the CARLA simulation environment, our method significantly outperforms vision-based baseline driving models on the challenging Bench2Drive benchmark in terms of route completion rate and driving score. After enhancement with our approach, the completion rates of the UniAD and VAD baselines reduced by 3.76% (23.0% relatively) and 4.54% (30.3% relatively), and the driving scores also increased by 4.26 (10.1% relatively) and 5.28 (12.5% relatively). In the supplementary material, we present several driving scenarios where the original models failed but could be completed successfully with the $E^3AD$ method, along with more detailed statistical results for further reference. These findings demonstrate the comprehensive capability and robust performance of $E^3AD$ in challenging environments.

## 4.4 Ablation Study and Comparison

In Tab. 3,we conducted ablation studies on key components of the Driving-Thinking model. When the Video Encoder was used solely as a pretrained visual feature extraction network without contrastive learning with LaBraM, experimental results showed that the additional visual features were redundant and did not yield performance improvements. Furthermore, when the Driving-Thinking model was unfrozen during the stage 2 training phase, the model tended to forget the acquired driving cognition during random training, which similarly did not lead to any improvement. These results comprehensively validate the effectiveness of the proposed contrastive learning paradigm, in which the Video Encoder learns from the features of LaBraM.

In addition, we conducted ablation studies on the sources of EEG data. Compared to novice drivers, using EEG from expert drivers significantly reduced the L2 error by bringing the model outputs closer to expert trajectories. Meanwhile, both expert and novice EEG enhanced the driving performance of the model by reducing collision rates. We also observed that, by mixing EEG data from both expert and novice drivers and training the Driving-Thinking model with a larger volume of EEG data, the driving performance of the model could be further improved.

In Tab. 4, we compared the three frameworks described in Section 3.3. Detailed experimental data on the design specifications and hyperparameter selection for each framework are provided in the supplementary material. Here, we present only the results of the optimal configurations. The first framework did not yield significant improvements, while both the second and third frameworks enhanced the model's driving ability by reducing collision rates and L2 error. Notably, the third framework achieved the greatest improvement in driving performance. These results demonstrate that the Driving-Thinking model learns high-level planning-related cognition, which can directly enhance the planning performance of driving models.

### 4.5 Analysis and Discussion

**What enhances E2E-AD?** Our ablation studies demonstrate that the improvements brought by $E^3AD$ to end-to-end autonomous driving baseline models do not originate from additional visual feature inputs, nor from the capacity of the Video Swin Transformer framework, but rather from the process of contrastive learning with LaBraM. LaBraM is a powerful brain-inspired large model capable of extracting general features from input EEG signals to accomplish complex and diverse downstream tasks. By learning on paired EEG and video data, the Driving-Thinking model can acquire corresponding driving cognition from the brain feature space based on visual inputs. Our experiments confirm that the information learned from the brain-inspired large model is what assists the driving model's planning, indicating that it is indeed driving cognition that benefits E2E-AD.

**How driving cognition enhances E2E-AD?** To investigate the mechanisms by which driving cognition enhances E2E-AD, we first formulated three hypotheses. Specifically, we hypothesized that the driving cognition learned by the Driving-Thinking model falls into three categories: visual feature attention, understanding, and decision reasoning. Each of these categories represents a hypothesis at an increasing level of cognitive difficulty and complexity. For each hypothesis, we carefully designed a corresponding framework, enabling the driving cognition associated with each hypothesis to enhance key planning features in an appropriate manner. Our experiments show that the effectiveness of the three frameworks in improving E2E-AD also increases correspondingly. This suggests that, for end-to-end driving tasks, the greatest improvements may result from high-level driving cognition, including knowledge related to human-like decision reasoning.

## 5 Contributions and Limitations

The $E^3AD$ paradigm is the first to incorporate human driving cognition to enhance end-to-end autonomous driving models. It can be directly applied to baseline models, achieving substantial improvements in driving performance with small computational cost, while reaching the level of state-of-the-art methods. Moreover, this study making novel contributions to the field of embodied human intelligence augmentation in AI algorithms. However, due to the cost and time of EEG acquisition and processing, the paired EEG–video dataset is relatively small. In addition, the precise mechanisms by which EEG alignment enhances planning have not been fully explored. To address these limitations, we commit to publicly releasing the dataset and data collection procedures soon to facilitate data expansion. Furthermore, our future work will investigate the models' underlying mechanisms.

## Acknowledgement

This work was supported by Beijing Natural Science Foundation L233033, the Beijing Municipal Science and Technology Project (Nos. Z231100010323005) and the China National Natural Science Foundation Youth Fund 62202267.

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

# A   Cognitive Data Collection

In this study, we recruited 27 participants; due to incomplete or corrupted data from some subjects, data from 20 participants were ultimately analyzed (10 novice drivers and 10 expert drivers). All experiments employed the Neuroscan 64-channel Quik-Cap[3] system and Headbox for EEG acquisition, with cables connecting the amplifier to the acquisition unit, the stimulator, and the headbox; stimulation and synchronization were performed using a YOGA stimuli host and E-Prime software. Data were recorded in real time via `CURRY 9`, and usage of experimental and analysis software was controlled by hardware dongles.

The experiment consisted of three phases: setup, recording, and termination. During setup, participants donned the electrode cap and applied conductive paste, the vehicle idled, and the in-car industrial PC and Docker environment were started; Dreamview was then launched to synchronize system time. During recording, the external operator configured the COM port in the tester software, `Curry 9` injected timestamps, and E-Prime triggered cross-modal synchronization via spacebar presses and audio cues (marker 161 for beeps, 192 for other events), while eye-tracking signals were monitored. At the end of the session, the vehicle parked and the parking brake was engaged, the operator stopped recording with `Ctrl+C`, verified completeness of the `.record` files, and saved the raw `.cnt` files.

Subsequent data processing involved electrode localization, removal of non-EEG electrodes (EKG, EMG, Trigger, etc.), re-referencing to M1/M2, band-pass filtering at $0.1$–$50$ Hz, bad-electrodes rejection, and artifact correction via ICA; events were then imported and data were epoched according to driving conditions. Using the processed data, we conducted statistical comparisons of fatigue levels, workload (theta/alpha plasticity), and task engagement between expert and novice drivers across driving scenarios, finding that experts exhibited more stable fatigue management and greater EEG adaptability in complex road sections.

Table A.1: EEG System Components and Accessories

| Type | Hardware Materials | Type | Hardware Materials |
|---|---|---|---|
| Headbox | 64-channel Quik-Cap, Headbox unit | Cables | Cable 1 (Amplifier → Acquisition), Cable 2 (Amplifier → Stimuli, white), Cable 3 (Amplifier → Headbox) |
| Master Unit 1 | Amplifier (wide-band main unit), Amplifier power cable | Software Licenses | Experimental software dongle, Analysis software dongle, E-Prime dongle |
| Master Unit 2 | Stimulus generator unit | Subject Consumables | EEG paste (bulk), paste syringe, abrasive gel, cotton swabs, adhesive tape, shampoo, hair dryer, disposable absorbent wipes |
| Master Unit 3 | Acquisition unit, Power cable | Other Equipment | Power strip, Expansion dock, Speaker set, Portable power bank, Serial-port interface box, Eye-tracking EEG sync cable |

# B   Hardware and Software

## B.1   Equipment

**Hardware**

- DJI Action 3 action cameras $\times 6$
- DJI Action battery charging case $\times 2$
- Insta360 X3 panoramic camera

---

[3]https://compumedicsneuroscan.com/products/caps/quik-cap/

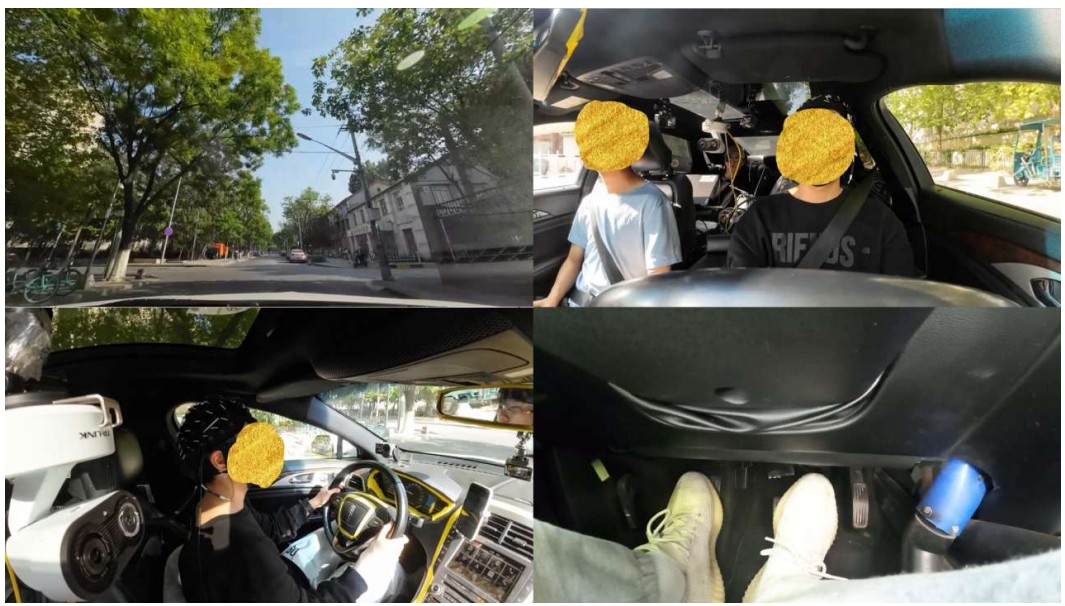

Figure A.1: Example snapshots of the video modalities: top left – Baseline (forward road view); top right – Driver1 (driver's face view); bottom left – Driver2 (driver's posture view); bottom right – Driver3 (driver's feet view).

- Insta360 charging case
- SD memory cards ×6
- Mounting brackets (multiple)
- High-power Bluetooth speaker (for audio synchronization)

**Software**

- E-Prime (for EEG synchronization)
- Jianying (video editing)

## B.2 Modalities

**RGB Action Cameras** Six in-vehicle viewpoints captured by DJI Action 3 cameras:

- **Baseline**: forward road view (dashcam)
- **Driver1**: driver's face
- **Driver2**: driver's feet
- **Driver3**: driver's posture
- **Passenger1**: front passenger posture
- **Passenger2**: rear passenger posture

Recording starts when synchronization is initiated and ends when the vehicle is safely parked.

**360° Panoramic** HDR 360° exterior view recorded by Insta360 X3, capturing surrounding road environment and vehicle pose. Recording interval matches the RGB cameras. All in-vehicle RGB action cameras (DJI Action 3) record at 1080 P resolution and 30 fps with a wide-angle lens ( 155°) and subsequent distortion correction; the panoramic camera (Insta360 X3) captures 360° surround video in 5.7 K HDR at 30 fps to ensure strict timestamp synchronization with the RGB cameras.

## B.3 Collection Procedure

1. **Preparation and Inspection**

1) Verify battery levels and free storage on all cameras; prepare spares.
2) Clean windshield and remove obstacles to minimize glare.
3) Mount and secure all six Action 3 cameras at predetermined positions; tighten brackets.
4) Check each camera's field of view against the setup diagram.
5) For the test drive, power on only the Baseline camera.
6) Configure Insta360 X3 to HDR mode and dual-lens (360°) capture; secure on roof mount and level the camera.

2. **Power-On**

1) During the practice drive, the external operator starts the Baseline camera.
2) Before the formal trial, the in-vehicle operator gives the start command; the external operator powers on all cameras.

3. **Recording Start**

1) In-vehicle operator issues a voice command to initiate recording on all RGB cameras simultaneously.
2) External operator confirms LED indicators and starts the Insta360 X3.

4. **Recording Stop**

1) Upon safe stop of the vehicle, the in-vehicle operator issues a voice command to stop all RGB cameras.
2) External operator stops the Insta360 X3 recording on command.

## C  Ablation Study and Comparison

First and foremost, it should be clarified that, in line with many recent studies[30, 18], we view L2 error solely as an indicator of model convergence, whereas the ultimate goal of autonomous driving is safe, collision-free operation. In this study, we doubled the coefficient of the L2 loss term in the source code of VAD-Base[12] and found that, although the L2 error decreased by 0.15 m (20.8%, respectively), the collision rate increased by 0.07% (31.8%, respectively), as shown in Table. A.2. Additionally, Ego-MLP[30] performs planning using only the vehicle's historical state information without any visual input, the L2 error achieved is comparable to that of VAD, yet both the collision rate and closed-loop evaluation results are terrible. These findings motivate us to adopt collision rate as the primary performance metric for open-loop evaluation during model design and hyperparameter selection, while relegating L2 error to a secondary role as an indicator of model convergence, and coefficient settings are kept consistent with the baseline model to ensure a fair comparison. Additionally, due to the low computational efficiency of UniAD[10], we primarily select VAD as the baseline model in the large-scale experiments discussed in the following sections.

As shown in Table. A.3, we compare the number of blocks, the dropout rate, and the number of cross-attention heads in the "Interact with the Ego Query" framework. Our observations indicate that, even with only one layer, the model achieves a stable improvement over the baseline. When the number of cross-attention heads is reduced by half, the model converges more easily and exhibits lower L2 error, while the collision rate remains unaffected. Additionally, dropout is a key parameter; excessively low dropout rates can lead to overfitting, resulting in degraded driving performance on the test set.

The comparison results of hyperparameters for the "Interact with Planning Features" framework are presented in Table.A.4. We observe that employing just one layer of our designed Decoder block can already lead to notable performance improvements. Nevertheless, appropriately increasing the number of layers can further enhance the overall performance of the model. It is also crucial to select a suitable dropout value, as excessively large or small dropout rates may lead to underfitting or overfitting, respectively, thereby degrading performance. Additionally, moderately reducing the number of attention heads can facilitate model convergence and improve driving performance. However, setting the number of attention heads too low may compromise the model's learning capacity.

For the "Attach to the Spatio-temporal Features" framework, we compare the impact of the number of vision feature queries $n_s$, which are selected by human visual attention. We observe that as $n_s$ increases, the model's L2 error gradually decreases. However, the collision rate does not decrease

accordingly and even increases when $n_s$ is excessively high, as shown in Table. A.5. This phenomenon can be attributed to the fact that this framework only captures visual features attended by the human brain, without learning planning-related driving cognition. These visual features primarily help the model to fit the expert driving trajectories, but their contribution to enhancing the model's driving performance is limited. Furthermore, when the visual features become overly redundant, they may even impair the model's performance.

Table. A.6 presents a comparison of detailed metrics between our model and baseline models under closed-loop evaluation. It can be observed that our model substantially reduces the rate of vehicle collisions and red light violations by 11.82% (38.8%, respectively) and 1.37% (37.6%, respectively). The rate of stop infraction is also significantly reduced by 0.46% (16.8%, respectively). In addition, occurrences of collisions with layouts and lane departures have also been mitigated. These results demonstrate that the $E^3AD$ paradigm achieves significant improvements in driving performance, particularly in complex driving tasks such as interacting with other vehicles and understanding traffic signals.

Table. A.7 presents the results of cross-dataset validation and generalization analysis. To further evaluate the robustness of our model in unseen geographic domains and to address the reviewer's concerns, we conducted additional experiments on the *nuScenes* dataset. The dataset was divided into two subsets according to the acquisition locations (Singapore and Boston). Specifically, the training set consists of 12,435 and 15,695 samples for Singapore and Boston, respectively, while the test set contains 2,929 and 3,090 samples. We trained the model using data collected from one location and evaluated it on the other, ensuring that the validation was performed on entirely unseen geographic domains. As shown in the Table. A.7, our model consistently outperforms the baseline (VAD-tiny) on unseen data. Particularly, when training on Singapore and validating on Boston, our approach achieves a 14.1% reduction in baseline L2 error and a 22.7% decrease in collision rate compared to the baseline.

Table A.2: Relationship Between L2 Error and Collision Rate.

| Method | L2(m)↓ | | | | Collision(%)↓ | | | |
|---|---|---|---|---|---|---|---|---|
| | 1s | 2s | 3s | Avg. | 1s | 2s | 3s | Avg. |
| VAD-Base[12] | 0.41 | 0.70 | 1.05 | 0.72 | 0.07 | 0.17 | 0.41 | 0.22 |
| VAD-Base (Doubling L2 loss) | 0.31 | 0.54 | 0.85 | 0.57 | 0.16 | 0.21 | 0.51 | 0.29 |
| Ego-MLP[30] | 0.46 | 0.76 | 1.12 | 0.78 | 0.21 | 0.35 | 0.58 | 0.38 |

Table A.3: Comparison of Hyperparameters in the "Interact with the Ego Query" Framework.

| Framework | Options | | | L2(m)↓ | | | | Collision(%)↓ | | | |
|---|---|---|---|---|---|---|---|---|---|---|---|
| | Layers | Dropout | heads | 1s | 2s | 3s | Avg. | 1s | 2s | 3s | Avg. |
| VAD[12] | _ | _ | _ | 0.41 | 0.70 | 1.05 | 0.72 | 0.07 | 0.17 | 0.41 | 0.22 |
| VAD (Reproduced) | _ | _ | _ | 0.40 | 0.70 | 1.04 | 0.71 | 0.10 | 0.16 | 0.44 | 0.23 |
| | **1** | 0.1 | 8 | 0.37 | 0.66 | 1.04 | 0.69 | 0.06 | 0.14 | 0.41 | 0.20 |
| | **2** | 0.1 | 8 | 0.39 | 0.68 | 1.03 | 0.70 | 0.09 | 0.19 | 0.42 | 0.23 |
| Ours | **4** | 0.1 | 8 | 0.36 | 0.62 | 0.95 | 0.64 | 0.08 | 0.19 | 0.41 | 0.23 |
| | 1 | **0.05** | 8 | 0.39 | 0.66 | 1.00 | 0.68 | 0.09 | 0.18 | 0.37 | 0.21 |
| | 1 | 0.1 | **4** | 0.33 | 0.58 | 0.92 | 0.61 | 0.07 | 0.14 | 0.39 | 0.20 |

# D   Visualization

Similar to other end-to-end methods[18, 10, 12, 16], we also provide visualization results, as shown in Fig. A.2. It demonstrates a successful case from the closed-loop simulation experiments: when a vehicle parked ahead on the right suddenly starts moving, the baseline model adopts a more aggressive and risky avoidance maneuver, attempting to pass quickly and causing a severe collision. In contrast, our model learns to interact with the suddenly moving vehicle by slowing down and waiting until it has completely departed before accelerating to proceed along the route, thus avoiding a collision.

Table A.4: Comparison of Hyperparameters in the "Interact with the Planning Features" Framework.

| Framework | Options | | | L2(m)↓ | | | | Collision(%)↓ | | | |
|---|---|---|---|---|---|---|---|---|---|---|---|
| | Layers | Dropout | heads | 1s | 2s | 3s | Avg. | 1s | 2s | 3s | Avg. |
| VAD | _ | _ | _ | 0.41 | 0.70 | 1.05 | 0.72 | 0.07 | 0.17 | 0.41 | 0.22 |
| VAD (Reproduced) | _ | _ | _ | 0.40 | 0.70 | 1.04 | 0.71 | 0.10 | 0.16 | 0.44 | 0.23 |
| | **4** | 0.1 | 8 | 0.33 | 0.59 | 0.93 | 0.62 | 0.06 | 0.14 | 0.41 | 0.20 |
| | **1** | 0.1 | 8 | 0.39 | 0.70 | 1.07 | 0.72 | 0.03 | 0.14 | 0.44 | 0.20 |
| | **2** | 0.1 | 8 | 0.34 | 0.59 | 0.91 | 0.61 | 0.12 | 0.20 | 0.32 | 0.21 |
| | **6** | 0.1 | 8 | 0.38 | 0.64 | 0.96 | 0.66 | 0.09 | 0.18 | 0.39 | 0.22 |
| Ours | 4 | **0.05** | 8 | 0.38 | 0.67 | 1.04 | 0.70 | 0.07 | 0.15 | 0.41 | 0.21 |
| | 4 | **0.15** | 8 | 0.34 | 0.61 | 0.96 | 0.63 | 0.06 | 0.17 | 0.40 | 0.21 |
| | 4 | 0.1 | **4** | 0.35 | 0.62 | 0.96 | 0.64 | 0.06 | 0.13 | 0.36 | 0.18 |
| | 4 | 0.1 | **2** | 0.33 | 0.59 | 0.93 | 0.62 | 0.06 | 0.14 | 0.41 | 0.20 |

Table A.5: Comparison of Hyperparameters in the "Attach to the Spatio-temporal Features " Framework.

| Framework | Options | L2(m)↓ | | | | Collision(%)↓ | | | |
|---|---|---|---|---|---|---|---|---|---|
| | $n_s$ | 1s | 2s | 3s | Avg. | 1s | 2s | 3s | Avg. |
| VAD | _ | 0.41 | 0.70 | 1.05 | 0.72 | 0.07 | 0.17 | 0.41 | 0.22 |
| VAD (Reproduced) | _ | 0.40 | 0.70 | 1.04 | 0.71 | 0.10 | 0.16 | 0.44 | 0.23 |
| | **4** | 0.38 | 0.67 | 1.04 | 0.70 | 0.09 | 0.17 | 0.400 | 0.22 |
| Ours | **8** | 0.38 | 0.67 | 1.02 | 0.69 | 0.09 | 0.17 | 0.38 | 0.21 |
| | **16** | 0.34 | 0.60 | 0.95 | 0.63 | 0.12 | 0.19 | 0.49 | 0.27 |

Table A.6: Comparison of $E^3AD$ and Baseline Models on Closed-Loop Metrics.

| Method | Closed-loop Metrics ↓ | | | | | |
|---|---|---|---|---|---|---|
| | Layouts Collision(%) | Pedestrians Collision(%) | Vehicles Collision(%) | Running Red light(%) | Stop Infraction(%) | Off-road(%) |
| VAD-Base | 15.46 | 0.91 | 30.46 | 3.64 | 2.73 | 23.64 |
| Ours | 15.00(↓3.1%) | 0.91 | 18.64(↓38.8%) | 2.27(↓37.6%) | 2.27(↓16.8%) | 22.73(↓3.8%) |

Table A.7: Cross-Dataset Validation and Generalization Study

| Method | L2 | | | | Col | | | |
|---|---|---|---|---|---|---|---|---|
| | 1s | 2s | 3s | Avg. | 1s | 2s | 3s | Avg. |
| Baseline (Training on Singapore) | 0.51 | 0.84 | 1.21 | 0.85 | 0.29 | 0.42 | 0.61 | 0.44 |
| $E^3AD$ (Training on Singapore) | 0.44 | 0.71 | 1.03 | 0.73 | 0.15 | 0.30 | 0.56 | 0.34 |
| Baseline (Training on Boston) | 0.45 | 0.77 | 1.17 | 0.80 | 0.22 | 0.35 | 0.76 | 0.44 |
| $E^3AD$ (Training on Boston) | 0.46 | 0.76 | 1.13 | 0.78 | 0.22 | 0.42 | 0.50 | 0.38 |

This case to some extent suggest that $E^3AD$ exhibits behaviors that are closer to human driving style and are characterized by enhanced safety.

Fig. A.3 illustrates the real-world data validation conducted in China. We collected real-world data using a vehicle-mounted platform equipped with an Insta360 panoramic fisheye camera and an inertial sensor. Following the nuScenes protocol, the panoramic videos were undistorted and split into six perspective-view videos. Using these multi-view videos together with CAN bus data, we evaluated our model. The $E^3AD$ model demonstrated smooth trajectory continuity.

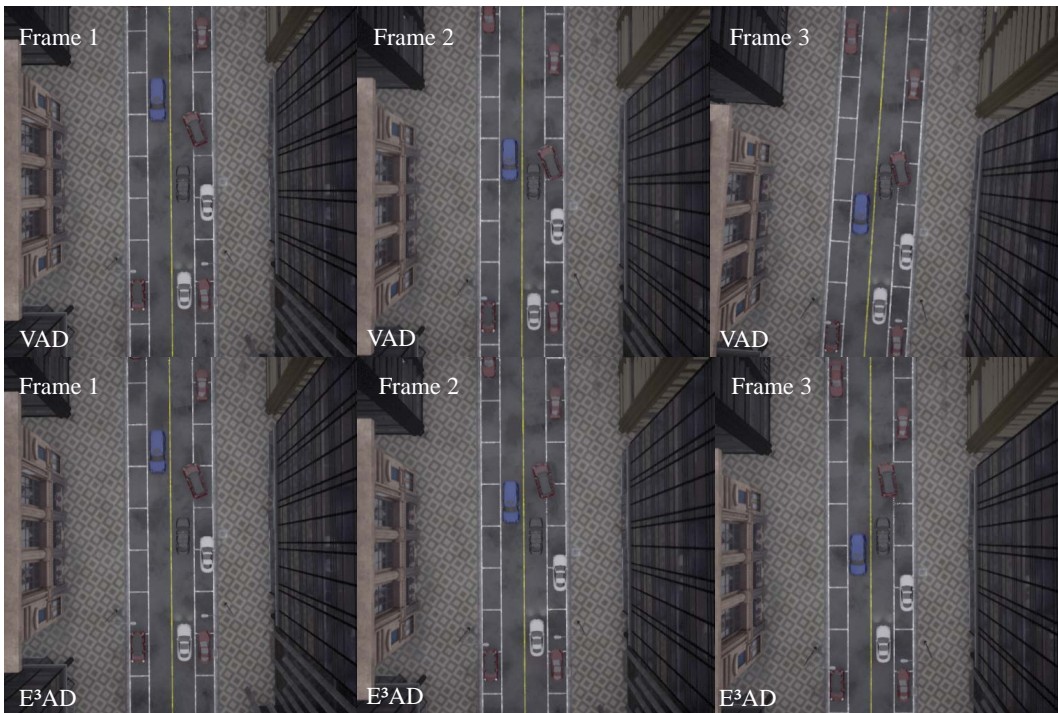

Figure A.2: Visualization Comparison of $E^3AD$ (VAD-Base) and the Baseline on Closed-loop Evaluation.

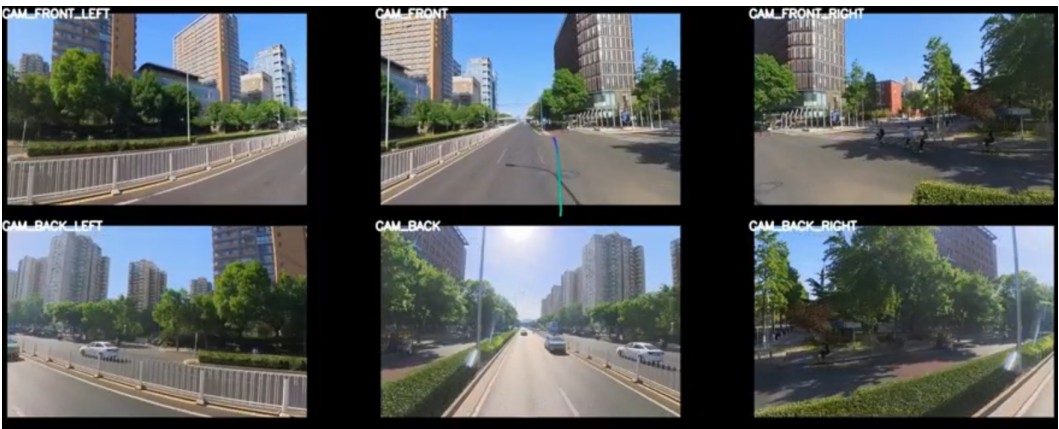

Figure A.3: Real-world data validation

