# OpenReview forum: "Embodied Cognition Augmented End2End Autonomous Driving"
_NeurIPS.cc/2025/Conference — NeurIPS 2025 poster_

### Official Review · Reviewer_bynr · 2025-06-29

**Clarity:** 3
**Significance:** 3
**Originality:** 3
**Rating:** 4
**Confidence:** 4

**Summary:**

This paper introduces 'Embodied Cognition Augmented End-to-End Autonomous Driving', a novel paradigm that integrates human cognitive features derived from EEG signals to enhance end-to-end autonomous driving planning. The method employs a two-stage training framework: In stage 1, it employs contrastive learning between a visual feature extraction network (Driving-Thinking Model) and a large EEG model (LaBraM) on a self-collected dataset. The dataset includes EEG signals from expert/novice drivers; Then, in stage 2, they integrate the frozen 'Driving-Thinking Model' into a popular end-to-end driving framework (e.g., VAD, UniAD) without EEG inputs. They evaluate on NuScenes (open-loop) and Bench2Drive (closed-loop), which show significant improvements.

**Questions:**

1. Could you visualize what the Driving-Thinking Model learns (e.g., saliency maps)? What specific cognitive aspects (e.g., hazard anticipation, path planning) do EEG features capture?

2. How do expert vs. novice cognitive features qualitatively differ?

**Ethical Concerns:**

["NO or VERY MINOR ethics concerns only"]

**Final Justification:**

The paper proposes a novel paradigm that integrates human cognitive features derived from EEG signals to enhance end-to-end autonomous driving planning. Authors have committed to releasing all datasets and collection details. This will provide insight for solving E2E driving. Considering the novelty and thorough ablations, I would like to improve my rate to borderline accept.

**Limitations:**

1. Reproducibility Risk:
The paper does not have a commitment to releasing the EEG dataset, despite its centrality to Stage 1. Code alone is insufficient without data.

2. Narrow Driver Demographics:
Data are collected from 20 male drivers (10 experts/10 novices), which limits generalizability across genders, ages, and cultural driving contexts.

3. Real-World Gaps:
The closed-loop is in CARLA simulations only. There are no real-world deployments or cross-dataset validation (e.g., Waymo).

**Paper Formatting Concerns:**

I do not have formatting concerns.

**Quality:**

3

**Strengths And Weaknesses:**

Strength:
1. Novelty: First work to incorporate human cognitive data (EEG) into end-to-end autonomous driving, pioneering a brain-inspired paradigm.
2. Rigorous Evaluation: Comprehensive testing across open-loop (L2 error, collision rate) and closed-loop (driving score, completion rate) metrics, with comparisons against SOTA baselines (VAD, UniAD).


Weakness:

1. Limited Alignment Validation in Stage 1:
1.1 There are no quantitative evaluations (e.g., similarity metrics) to verify if Stage 1 contrastive learning successfully aligns visual and EEG features.
1.2 It is unclear how well cognitive features correlate with driving semantics beyond collision-rate improvements. Does it have more insights or theoretical analysis?

2. Theoretical Gaps:
2.1 The paper lacks a mechanistic analysis of why EEG features improve planning (e.g., hazard anticipation, decision-making patterns). There is no discussion on cognitive feature interpretability (e.g., attention maps linking EEG to critical driving events).

3. Dataset Scalability & Release:
The collected EEG dataset is the core contribution of the paper. However, the collected EEG dataset is small (20 drivers, 2.3k clips), with no mention of public release. This hinders reproducibility and community validation.
I also have scalability concerns. Could the EEG data be easily scaled up? Recently, the autonomous driving community is using larger and larger data to scale the performance. However, from the paper, I could not figure out if this method could still achieve better performance when scaling data. If so, could it solve various corner cases with scaled data?

4. Overreliance on LaBraM:
The paper uses a frozen LaBraM model pre-trained on thousands of hours of EEG data. No ablation on LaBraM’s necessity or alternatives for resource-constrained settings.

---

> ### Author Rebuttal · Authors · 2025-07-31
>
> Thank you for highlighting the innovation of our integration of human cognitive data (EEG) into end-to-end autonomous driving, as well as recognizing the rigor of our experimental design. We appreciate your concerns and address each of them below.
>
> ---
> ## Response to Weakness
> **Weakness 1:**
> >Regarding point 1.1, since there are no standard metrics for EEG-video alignment, we adopt the widely used contrastive loss from CLIP-related works as a proxy. Our loss is a symmetric cross-entropy over the similarity matrix of normalized EEG and video embeddings in a minibatch. The contrastive loss encourages higher cosine similarity for matching EEG-video pairs than mismatched ones. In our experiments, the evaluation contrastive loss dropped from 2.7 to 2.2, compared to a random baseline of roughly ln(16) ≈ 2.77, indicating stronger alignment despite dataset noise.
>
> >Regarding point 1.2, our improvements extend beyond standard metrics like L2 error and collision rate. Supplementary Table 6 compares detailed driving metrics under the large-scale Bench2Drive benchmark (220 routes, 12 towns, 23 weather conditions in CARLA v2). Our model reduces vehicle collisions and red light violations by 11.82% (38.8%) and 1.37% (37.6%), respectively, and decreases stop infractions by 0.46% (16.8%). Collisions with layouts and lane departures are also mitigated. These results show that **$E^3AD$** significantly enhances driving performance, especially in complex scenarios involving traffic interactions, mainly through a more disciplined driving style and collision avoidance.
>
> ---
>
> **Weakness 2:**
>
> >Thank you for pointing this out. In the current version of our work, We have conducted preliminary experiments exploring cognitive feature-enhanced driving; however, we have not yet performed a detailed investigation into the underlying mechanisms. Therefore, we plan to visualize the outputs of our Driving-Thinking Model and explore their relationship with the visual driving scene, in order to better understand the cognitive-planning connection.
>
> >It is worth noting that our bold attempt to integrate cognitive features into autonomous driving is grounded in the theoretical foundations of this field. Prior research has shown that specific EEG components—such as the P400, associated with attention engagement [1], and the N500, associated with unpredictability processing [2]—are characteristic markers of a driver’s hazard detection in complex scenarios [3]. We believe these neural signals can play a crucial role in enhancing planning capabilities when integrated into autonomous driving models. To provide more intuition behind our design, we will include additional discussion on these neuroscience findings in the revised version.
>
> **References**
> **(1)** The cortical development of specialized face processing in infancy
> **(2)** Predicting outcomes of decisions in the brain
> **(3)** Annotating Covert Hazardous Driving Scenarios Online: Utilizing Drivers’ Electroencephalography (EEG) Signals
>
> ---
>
> **Weakness 3:**
>
> Thank you for highlighting these important concerns.
> >First, We are committed to releasing the full paired EEG-video dataset, ensuring its usability and clarity, along with detailed usage instructions and ethical safeguards.
>
> >Second, Regarding scalability, we commit to publicly sharing the data collection process and expanding the dataset, with plans to release it within the next year. We acknowledge the challenges of scaling EEG data collection due to the need for specialized equipment. Therefore, we are actively seeking collaborations with industry partners.
>
> >Thirdly, with respect to dataset size, in this work we leverage LaBraM to enhance the generalization capability of our model. Specifically, we freeze LaBraM during training and use contrastive learning in the proposed Driving–Thinking model to learn generalizable features from LaBraM. This allows us to mitigate the limitations arising from the relatively small size of our collected dataset, thanks to LaBraM’s strong generalization and the universality of its features.
>
> ---
>
> **Weakness 4:**
>
> >Thank you for your thoughtful review and question. Interestingly, we have explored the alternative you mentioned: in our initial work, a pre-trained ResNet50 and projection layer were used to align visual and cognitive representations, with MSE loss to reconstruct EEG signals. This approach yielded notable improvements over the baseline, reducing average L2 error by 0.105 m (13.9%) and collision rate by 0.03% (13.6%), as shown in the table below. However, we later found that combining LaBraM with contrastive learning offered significantly better generalization—especially for collision rate and closed-loop scenarios—so we adopted this improved method. We commit to providing these results and code in the supplementary materials of the revised version.
>
> | Model | L2 (1s) | L2 (2s) | L2 (3s) | L2 (Avg) | Col (1s) | Col (2s) | Col (3s) | Col (Avg) |
> |------------------------------:|--------:|--------:|--------:|---------:|---------:|---------:|---------:|----------:|
> | Baseline (VAD)|    0.41 |0.70 | 1.05 |0.72 |0.07 |     0.17 |     0.41 |      0.22 |
> |**$E^3AD$** (regression) – all|    0.33 |    0.60 |    0.92 |     0.62 |     0.09 |     0.15 |     0.33 |      0.19 |
> |**$E^3AD$**(regression) – expert    |    0.34 |    0.60 |    0.93 |     0.62 |     0.11 |     0.17 |     0.34 |      0.21 |
> |**$E^3AD$**(regression) – novice    |    0.40 |    0.65 |    0.96 |     0.67 |     0.11 |     0.20 |     0.35 |      0.22 |
>
> > Regarding resource-constrained settings: Our Driving–Thinking Model is trained once using paired EEG and video data with contrastive learning (LaBraM). It can then be integrated into various end-to-end autonomous driving models via the **${E^3AD}$** framework without retraining. As shown in Table 1 of manuscript, we measured FPS before and after **${E^3AD}$** integration on the same hardware (NVIDIA A100 GPU). Results show FPS remained stable for both lightweight models like VAD-tiny and larger models like Uniad. Therefore, our method adds minimal computational overhead and suits resource-limited environments.
>
> ---
>
> ## Responses to Specific Questions
>
> **Question 1:**
> >First, as discussed in the Response to Weakness 2, previous research has demonstrated that specific EEG signals are associated with drivers' risk perception and unpredictability processing. These include a driver’s perception of scene hazards and their planning strategies. The experiments in our paper also support that EEG signals primarily encode information related to planning styles in different driving scenarios.
>
> >Second, we agree that visualizing the learned features—particularly in relation to specific cognitive aspects—would provide valuable insights. Recently, we have attempted to perform clustering analysis on the output features of the Driving-Thinking Model in different scenarios. However, due to the limited time available for this rebuttal, we commit to discussing these results in the revised version.
>
> **Question 2:**
> > Before we designed our model, we visualized the EEG PSD (power spectral density) topomap of expert group and novice group in the frequency model (we use python mne package for visualization). We split five frequency bands (i.e. Delta, theta, alpha, beta and gamma) and find expert drivers' power concentrate on occipital lobe (visual areas) while novice drivers' power concentrate on frontal lobe (higher cognitive functions, e.g. Decision making). That's an interesting finding. It seems that novice needs to pay more energy to think about what I should do next, while expert only needs to see, and he knows how to do next.
>
> ---
>
> ## Response to Limitations
>
> **Limitation 1:**
> >Please refer to Response to weakness 3
>
> **Limitation 2:**
> >For this limitation, we will clearly state it in the Limitation section. In addition, as addressed in our response to Weakness 3, we will make the dataset and the corresponding data collection procedures publicly available. On this basis, we will further expand the dataset to better address this limitation.
>
> **Limitation 3:**
> >To address this limitation, we took the following steps:
>
> >First, we collected real-world data using a vehicle-mounted platform equipped with an Insta360 panoramic fisheye camera and an inertial sensor. The panoramic videos were undistorted and split into six view-angle videos, following the nuScenes protocol. Using these multi-view videos along with CAN bus data, we evaluated both the baseline model trained on nuScenes and our **$E^3AD$** model. The **$E^3AD$** model showed better trajectory continuity and more accurate turning intention recognition. We will provide these videos as supplementary material.
>
> >Second, to verify cross-dataset performance and generalization to unseen scenarios, we conducted experiments on the nuScenes dataset divided by location: Singapore with 12,435 training and 2,929 test samples, and Boston with 15,695 training and 3,090 test samples. The model was trained on one city’s data and tested on the other, ensuring evaluation on truly unseen geographic domains. As shown in the table below, our model demonstrates a clear advantage over the baseline (VAD-tiny) in validation on unseen data.
>
> | Scenario | L2 (1s) | L2 (2s) | L2 (3s) | L2 (Avg) | Col (1s) | Col (2s) | Col (3s) | Col (Avg) |
> |---------------------------------------------|--------:|--------:|--------:|---------:|---------:|---------:|---------:|----------:|
> | Baseline (Training on Singapore)| 0.51 |    0.84 |    1.21 |     0.85 |     0.29 | 0.42 | 0.61 | 0.44 |
> | **$E^3AD$** (Training on Singapore)  | 0.44 |    0.71 |    1.03 |     0.73 |     0.15 |     0.30 | 0.56 |  0.34 |
> | Baseline (Training on Boston)  | 0.45 | 0.77 |    1.17 |     0.80 |     0.22 |     0.35 |     0.76 |      0.44 |
> | **$E^3AD$** (Training on Boston)| 0.46 | 0.76 |    1.13 |     0.78 |     0.22 |     0.42 |     0.50 |      0.38 |

---

> > ### Author Response · Authors · 2025-08-07
> >
> > Dear Reviewer,
> >
> > Thank you for acknowledging the innovation and experimental rigor of our work. We have addressed your concerns with careful and detailed responses:
> >
> > - **Weakness 3 and Limitation 1** regarding the dataset release:
> >   We have committed to making the dataset and its collection methodology publicly available. We have also provided further details about how we leverage LaBraM to enhance the generalization capability of our model.
> >
> > - **Weakness 1 and Question 1**:
> >   We clarified that the relevant visualization results have already been included in the supplementary material. We will further provide visualizations of open-loop evaluation and real vehicle data validation in the revised manuscript.
> >
> > - **Weakness 4 and Limitation 3**:
> >   We have supplied additional experimental results and conducted further experiments, and we are confident that these efforts effectively address your concerns.
> >
> > - **Weakness 2**:
> >   We have provided more theoretical justification and included corresponding references for your review.
> >
> > - **Limitation 2**:
> >   We acknowledge this and reaffirm our commitment to releasing the dataset, with a statement to be included in the revised manuscript's Limitations section.
> >
> > We hope that these clarifications and the newly added experiments address your concerns. If you are satisfied, we kindly request you to consider updating the score to reflect the newly added results and discussion. We remain committed to addressing any further questions you may have during the discussion phase.
> >
> > Best wishes.

---

### Official Review · Reviewer_7cU7 · 2025-07-01

**Clarity:** 3
**Significance:** 2
**Originality:** 3
**Rating:** 4
**Confidence:** 4

**Summary:**

The paper proposes an end-to-end model performance enhancement method based on contrastive learning. By integrating human driving cognitive information to train the “Driving Thinking Model” as an additional feature extractor, the Planning ability of the end-to-end model is enhanced.

**Questions:**

1. For Weakness 1, could you provide efficiency experiments to show how much computational overhead is added after applying the proposed method?
2. What is the basis for using EEG as a representative of human driving cognitive information? Have other alternatives been considered?
3. According to Figure 1, the proposed method requires modifications to the original model and additional adaptation training. Is there an analysis of these extra costs?
4. The improvement of the proposed method on VAD-BASE and UniAD is impressive. However, these works are relatively old. Could you demonstrate the improvement effects on more recent end-to-end autonomous driving models? If so, I would consider increasing the score.

**Ethical Concerns:**

["NO or VERY MINOR ethics concerns only"]

**Final Justification:**

Most of my concerns have been well addressed. Thus, I am leaning to my current positive ratings.

**Limitations:**

Yes.

**Paper Formatting Concerns:**

No.

**Quality:**

2

**Strengths And Weaknesses:**

Strengths:
1. It presents the concept of “Driving-Thinking Model”, which uses contrastive learning to integrate human driving cognition for effective feature extraction.
2. The Driving-Thinking Model, once trained, serves as an extra feature extractor, augmenting the existing end-to-end model's planning capability.
3. The authors have collected a dataset comprising EEG and driving information, contributing to future autonomous driving research.

Weaknesses:
1. The claim that the method enhances the baseline end-to-end autonomous driving model with minimal computational overhead lacks experimental evidence.
2. Using EEG as a representation of human driving cognition to train the model to integrate human perception into environmental feature extraction is questionable in terms of fully capturing human driving intelligence. (See Question 2)
3. The baselines used for comparison in the experiments are relatively outdated. (See Question 4)

---

> ### Author Rebuttal · Authors · 2025-07-31
>
> Thank you for recognizing the innovation of our proposed approach, its effectiveness, and the contributions we have made to the field of autonomous driving. We appreciate your concerns and will provide detailed, point-by-point responses to each of them below.
>
> ---
>
> ## Response to Weakness
> **Weakness 1:**
> > Thank you for your careful review of our manuscript. However, regarding this weakness, we would like to clarify the following: In the last column of Table 1 in the manuscript, we provide the Frames Per Second (FPS) for each model. FPS is a commonly used metric to characterize the computational efficiency and real-time performance of autonomous driving models. In Table 1, for all end-to-end models incorporated into the **${E^3AD}$** method, we recorded FPS both before and after integration on the same hardware setup (NVIDIA A100 GPU). We observed that after integration with **${E^3AD}$**, the FPS remains relatively stable for all models, whether it is a high-efficiency model like VAD-tiny or a large model like Uniad. Based on this observation, we concluded that the method enhances the baseline end-to-end autonomous driving models with minimal computational overhead.
>
> ---
>
> **Weakness 2:**
> Thank you for your question.
>
> >First, it is worth noting that our bold attempt to integrate cognitive features into autonomous driving is grounded in the theoretical foundations of this field. Prior research has shown that specific EEG components—such as the P400, associated with attention engagement [1], and the N500, associated with unpredictability processing [2]—are characteristic markers of a driver’s hazard detection in complex scenarios [3]. We believe these neural signals can play a crucial role in enhancing planning capabilities when integrated into autonomous driving models. To provide more intuition behind our design, we will include additional discussion on these neuroscience findings in the revised version.
>
> >Second, most existing autonomous driving datasets (e.g., nuScenes, Waymo) only provide supervision from driver trajectories and CAN bus data, overlooking drivers’ internal cognitive processes before taking action. While recent approaches use visual-language or large language models to approximate human reasoning, these methods remain indirect and are not suitable for the sub-second decision-making required in real driving. In contrast, EEG signals offer a more direct and real-time measure of drivers' cognitive states, capturing otherwise unobservable thoughts and risk assessments. Therefore, our intention is not to model all aspects of human driving intelligence through EEG, but rather to extract those direct, real-time cognitive features that are most relevant to driving.
>
> **References**
> **(1)** The cortical development of specialized face processing in infancy
> **(2)** Predicting outcomes of decisions in the brain
> **(3)** Annotating Covert Hazardous Driving Scenarios Online: Utilizing Drivers’ Electroencephalography (EEG) Signals
>
> ---
>
> **Weakness 3:**
>
> >Thank you for your valuable suggestion. We agree that demonstrating the improvement effects on more recent end-to-end autonomous driving models would further strengthen the persuasiveness of our study. To this end, we have made the following efforts for your consideration:
>
> >First, we conducted experiments using GenAD as a baseline model. It is important to note that we observed a discrepancy between the official checkpoints provided by GenAD and the performance reported in their original paper. Specifically, while the L2 error is slightly lower, the collision rate is substantially higher. We then reproduced their results using the official configuration on eight NVIDIA A100 GPUs and obtained similar findings. Therefore, our experiments reported are based on the official checkpoints. After integrating **${E^3AD}$** into GenAD, the performance of our model shows a significant improvement compared to the reproduced results of the baseline, and it also surpasses the official checkpoint. We would like to clarify that, due to the limited time available for the rebuttal, we reduced the total number of training epochs from 60 to 12. We expect that using the original number of training epochs would yield even more promising results.. These experiments further validate the generalization capability of our method.
>
> | Model              | L2 (1s) | L2 (2s) | L2 (3s) | L2 (Avg) | Col (1s) | Col (2s) | Col (3s) | Col (Avg) |
> |--------------------|--------:|--------:|--------:|---------:|---------:|---------:|---------:|----------:|
> | GenAD (official)   |   0.25  |   0.46  |   0.76  |    0.49  |    0.11  |    0.21  |    0.45  |     0.26  |
> | GenAD (reproduce)  |   0.25  |   0.46  |   0.76  |    0.49  |    0.14  |    0.26  |    0.50  |     0.30  |
> | **${E^3AD}$** (GenAD)        |   0.24  |   0.44  |   0.74  |    0.47  |    0.10  |    0.21  |    0.42  |     0.24  |
>
> >Second, due to the limited rebuttal period, we commit to making further efforts to explore other recent end-to-end autonomous driving models (such as LAW) and recent benchmarks (such as NavSim), while continuing to enhance the performance of our method on GenAD. To demonstrate our ongoing commitment to improving the model, we also provide in the table below our recent efforts applying **${E^3AD}$**  to VAD. Through the expansion of EEG data and model optimization, our model’s performance has achieved a higher level. We further commit that these efforts will be fully documented in the main text or supplementary materials of the final version of the manuscript.
>
> | Model         | L2 (1s) | L2 (2s) | L2 (3s) | L2 (Avg) | Col (1s) | Col (2s) | Col (3s) | Col (Avg) |
> |---------------|--------:|--------:|--------:|---------:|---------:|---------:|---------:|----------:|
> | Baseline (VAD) |  0.41   |  0.70   |  1.05   |   0.72   |   0.07   |   0.17   |   0.41   |    0.22   |
> | **${E^3AD}$** (VAD)      |  0.40   |  0.68   |  1.03   |   0.70   |   0.04   |   0.12   |   0.30   |    0.15   |
>
> ## Responses to Specific Questions
>
> **Question 1:**
>  >Please refer to Response to weakness 1
>
> **Question 2:**
> > For the first question, please refer to our Response to Weakness 2. Regarding the second question, we have not yet seen relevant reports addressing the relationship between other physiological signals and driving. However, we believe that future studies could explore this topic using data from functional near-infrared spectroscopy (fNIRS) and magnetoencephalography (MEG).
>
> **Question 3:**
> Thank you for your question and for raising this detailed point.
> >Our proposed Driving–Thinking Model is trained with paired EEG and video data using contrastive learning based on LaBraM. It learns general-purpose representations from LaBraM that can be directly applied to different end-to-end autonomous driving models. For a given set of EEG data, the Driving–Thinking Model only needs to be trained once. Thereafter, it can be seamlessly integrated into any end-to-end model via the **${E^3AD}$** framework and trained according to the original procedure and schedule of that specific model. Regarding the training cost of the Driving–Thinking Model, we provide the following data for your reference: 3 hours of training on a single NVIDIA A100 GPU with 10GB VRAM.
>
> **Question 4:**
> >Thank you for your valuable feedback. We have made further efforts and attempts in response. Please refer to our Response to Weakness 3 for details.
> ---

---

### Official Review · Reviewer_w1pE · 2025-07-02

**Clarity:** 2
**Significance:** 2
**Originality:** 2
**Rating:** 4
**Confidence:** 3

**Summary:**

This paper introduces E3AD, a novel paradigm that explicitly incorporates human driving cognition into E2E autonomous-driving planning. The core idea is to perform contrastive learning between a visual feature-extractor and a large-scale EEG model so that latent human cognitive signals can be distilled and injected into the driving policy. The authors collect a dedicated cognitive dataset to support this procedure and adopt popular E2E driving models as baselines on publicly available benchmarks. Both open-loop and closed-loop experiments are conducted, followed by ablation studies, to demonstrate that E3AD consistently improves the baseline planners and to quantify the contributions of the driving cognition and the contrastive-learning strategy.

**Questions:**

Please refer to weakness

**Ethical Concerns:**

["NO or VERY MINOR ethics concerns only"]

**Final Justification:**

The visualizations clearly addressed my earlier concerns, and the new experiments resolved the remaining issues I had. Consequently, I am changing my recommendation from Borderline Reject to Borderline Accept.

**Limitations:**

yes

**Quality:**

2

**Strengths And Weaknesses:**

Strengths Contribution
1. Conceptual novelty. This is the first work that fuses human driving cognition with an E2E driving model, effectively bridging embodied intelligence and conventional deep-learning pipelines.
2. Methodological soundness. The proposed contrastive-learning–based Driving Thinking Model allows the network to infer cognitively relevant information directly from raw visual input.
3. Efficiency. The cognitive branch introduces negligible computational overhead yet yields measurable gains in both open- and closed-loop planning metrics.

Limitations Weakness
1. Lack of qualitative evidence. Beyond numerical improvements, please provide intuitive visualizations (e.g., side-by-side roll-outs of identical scenarios) showing that the cognitive module prevents collisions that the vanilla model fails to avoid.
2. Generality study. Given that GenAD outperforms the baseline models listed in Table 1, integrating E3AD into GenAD—or, more broadly, into every listed backbone—would make the empirical case far more compelling.
3. Presentation. Add an explicit “Contributions” paragraph at the end of the Introduction to clarify the paper’s key contributions. Highlight the best-performing values in all result tables in bold rather than with asterisks to enhance readability.

---

> ### Author Rebuttal · Authors · 2025-07-31
>
> Thank you for recognizing the innovation of our method and for acknowledging that our work effectively bridges embodied intelligence and conventional deep-learning pipelines. We also appreciate your positive comments on the methodological soundness and our results. We are grateful for your feedback and will provide detailed, point-by-point responses to each of them below.
>
> ---
>
> ## Response to Weakness 1
>
> >We agree that visualization results provide an intuitive demonstration. In fact, the supplementary materials already contain the visualization results you requested, which were omitted from the main text due to space limitations. In Figure 2 of the supplementary, we present a visualization comparison between **$E^3AD$(VAD-Base)** and the baseline under closed-loop evaluation. This figure illustrates a successful case from the closed-loop simulation experiments: when a vehicle parked ahead on the right suddenly starts moving, the baseline model responds with a more aggressive and risky avoidance maneuver, attempting to pass quickly and consequently causing a severe collision. In contrast, our model exhibits a more cautious behavior by slowing down and waiting until the moving vehicle has fully cleared the path before accelerating to continue along the route, thereby avoiding a collision. This case suggests, to some extent, that **$E^3AD$**  demonstrates driving behaviors more consistent with human driving style and improved safety characteristics.
>
> >In addition, in response to your valuable suggestions, we commit to providing additional visual results in the revised version—including videos of our real-world data validation and visualization comparisons from open-loop evaluations—to more intuitively demonstrate the performance of our model. For the videos of real-world data validation, we collected real-world data using a vehicle-mounted platform equipped with an Insta360 panoramic fisheye camera and an inertial sensor. The panoramic videos were undistorted and split into six view-angle videos, following the nuScenes protocol. Using these multi-view videos along with CAN bus data, we evaluated both the baseline model trained on nuScenes and our **$E^3AD$** model. The **$E^3AD$** model showed better trajectory continuity and more accurate turning intention recognition.
>
> ---
>
> ## Response to Weakness 2
>
> Thank you for your valuable suggestion. We agree that additional generality studies would make the empirical evidence more compelling. In response, we have made the following efforts for your consideration:
> > First, we conducted experiments using GenAD as a baseline model. It is important to note that we observed a discrepancy between the official checkpoints provided by GenAD and the performance reported in their original paper. Specifically, while the L2 error is slightly lower, the collision rate is substantially higher. We then reproduced their results using the official configuration on eight NVIDIA A100 GPUs and obtained similar findings. Therefore, our experiments reported are based on the official checkpoints. After integrating **${E^3AD}$** into GenAD, the performance of our model shows a significant improvement compared to the reproduced results of the baseline, and it also surpasses the official checkpoint. We would like to clarify that, due to the limited time available for the rebuttal, we reduced the total number of training epochs from 60 to 12. We expect that using the original number of training epochs would yield even more promising results.. These experiments further validate the generalization capability of our method.
>
> | Model             | L2 (1s) | L2 (2s) | L2 (3s) | L2 (Avg) | Col (1s) | Col (2s) | Col (3s) | Col (Avg) |
> |-------------------|--------:|--------:|--------:|---------:|---------:|---------:|---------:|----------:|
> | GenAD (official)  |    0.25 |    0.46 |    0.76 |     0.49 |     0.11 |     0.21 |     0.45 |      0.26 |
> | GenAD (reproduce) |    0.25 |    0.46 |    0.76 |     0.49 |     0.14 |     0.26 |     0.50 |      0.30 |
> | **$E^3AD$** (GenAD)       |    0.24 |    0.44 |    0.74 |     0.47 |     0.10 |     0.21 |     0.42 |      0.24 |
>
> >Second, due to the limited rebuttal period, we commit to making further efforts to explore other recent end-to-end autonomous driving models (such as LAW) and recent benchmarks (such as NavSim), while continuing to enhance the performance of our method on GenAD. To demonstrate our ongoing commitment to improving the model, we also provide in the table below our recent efforts applying **${E^3AD}$**  to VAD. Through the expansion of EEG data and model optimization, our model’s performance has achieved a higher level. We further commit that these efforts will be fully documented in the main text or supplementary materials of the final version of the manuscript.
>
> | Model          | L2 (1s) | L2 (2s) | L2 (3s) | L2 (Avg) | Col (1s) | Col (2s) | Col (3s) | Col (Avg) |
> |---------------:|--------:|--------:|--------:|---------:|---------:|---------:|---------:|----------:|
> | Baseline (VAD) |    0.41 |    0.70 |    1.05 |     0.72 |     0.07 |     0.17 |     0.41 |      0.22 |
> | **$E^3AD$** (VAD)      |    0.40 |    0.68 |    1.03 |     0.70 |     0.04 |     0.12 |     0.30 |      0.15 |
>
> ---
>
> ## Response to Weakness 3
>
> > Thank you for your valuable suggestions regarding the writing. Following your advice, we have replaced the "*" symbols in the result tables within the manuscript with bold font to highlight the best-performing values. Additionally, we have added an explicit "Contributions" paragraph at the end of the introduction section, stating “The $E^3AD$ paradigm is the first to incorporate human driving cognition to enhance end-to-end autonomous driving models, yielding significant findings. $E^3AD$ can be directly applied to baseline end-to-end driving models, achieving substantial improvements in driving performance with only a tiny increase in computational cost, while reaching the level of state-of-the-art methods. Moreover, our study investigates the methods and underlying mechanisms by which driving cognition enhances end-to-end planning, making novel contributions to the field of embodied human intelligence augmentation in AI algorithms. At the same time, Our work represents an exploration of a more end-to-end styled autonomous driving framework, enabling the model to acquire richer semantic information from raw data through implicit supervision, rather than being limited to manually annotated labels.”
>
> We sincerely appreciate your practical recommendations, which have greatly improved the readability of our manuscript.

---

> > ### Comment · Reviewer_w1pE · 2025-08-05
> >
> > The rebuttal clearly addressed my earlier concerns, and the new experiments resolved the remaining issues I had. Consequently, I am changing my recommendation from Borderline Reject to Borderline Accept.

---

### Official Review · Reviewer_cnpF · 2025-07-04

**Clarity:** 3
**Significance:** 3
**Originality:** 4
**Rating:** 4
**Confidence:** 4

**Summary:**

The authors presents EAD, a novel framework that enhances vision-only driving models by incorporating human EEG-derived cognitive features. In the first stage,  a Video‑Swin backbone is trained on synchronized in-car video and 64‑channel EEG data collected from 27 drivers. Through contrastive learning, the video embeddings are aligned with features extracted by the large EEG model LaBraM, resulting in a vision network that implicitly captures human driving cognition. In the second stage, the frozen Driving‑Thinking model is integrated into end‑to‑end planning modules such as VAD and UniAD using three approaches: attaching to BEV features, enriching ego-queries, and refining planning features. Evaluated on both NuScenes open-loop and Bench2Drive closed-loop benchmarks, EAD demonstrates a reduction in collision rates with no significant impact on inference speed.

**Questions:**

1. Will you release the paired EEG‑video dataset and pretrained Driving‑Thinking model? If so, under what licence and with what privacy safeguards?

2.  Have you tested EAD on unseen geographic domains (e.g., Waymo Open, CARLA Town10) or with synthetic EEG noise to evaluate robustness?

3. How does performance scale if only 5, 10, 15 subjects are used to train the contrastive stage?

**Ethical Concerns:**

["NO or VERY MINOR ethics concerns only"]

**Limitations:**

1. The generalizability is yet to be explored.
2. More diverse geographic domains needs to be validated in the future.

**Paper Formatting Concerns:**

The paper is easy to follow.

**Quality:**

3

**Strengths And Weaknesses:**

Strengths:

1. First to fuse EEG cognition with end‑to‑end planning; contrastive loss is simple yet effective.

2. Clever design means no additional sensors are required in deployment.

3. The authors test three hypotheses (attention, ego‑query guidance, decision reasoning) and show that the planning‑feature interaction yields the biggest gain.

Weaknesses:

1. A small dataset collected and used by the authors raises concerns about over‑fitting and model bias. A broader demographic coverage analysis would strengthen claims.

2. Brain data are sensitive. A dedicated limitations subsection is needed (data consent, anonymisation, potential misuse).

---

> ### Author Rebuttal · Authors · 2025-07-31
>
> Thank you for recognizing the novelty of our work, appreciating the creativity of our design, and acknowledging our contributions. We appreciate your concerns and will provide detailed, point-by-point responses to each of them below.
>
> ---
> ## Response to Weakness 1
>
> >Thank you for your question. We agree that overfitting is a common concern when working with small datasets. To mitigate this, we fine-tune our model through a contrastive learning process with LaBraM, a large-scale EEG foundation model pre-trained on over 2,000 hours of EEG recordings. We believe this design can reduce the risk of overfitting by providing robust and generalizable feature representations.
>
> >In addition, we monitored the contrastive loss of our Driving-Thinking Model during Stage 1 training. We observed that the losses on both the training and evaluation sets converge smoothly and consistently, which further suggests that overfitting is not a major issue under our current experimental setting. We will provide the experimental evidence for this part in the supplementary materials of the revised manuscript.
>
> ---
>
> ## Response to Weakness 2
>
> >Thank you for raising this important concern. We confirm that the entire data collection process strictly followed legal and ethical guidelines, and was conducted under the supervision and approval of our host institution. All necessary permissions and consents were obtained prior to data acquisition.
>
> >We acknowledge the sensitivity of brain data and fully agree that a dedicated limitations subsection is necessary. In the final version of the paper, we will include a clear statement detailing data consent procedures, anonymization protocols, and a discussion of potential risks and safeguards to prevent misuse of cognitive data.
>
> ---
>
> ## Responses to Questions
>
> **Question 1:**
> > Thank you for your question. We are committed to releasing our paired EEG-video dataset and the pre-trained Driving-Thinking model to support future research in this area. Currently, we are carefully analyzing, organizing, and documenting the dataset to ensure it is accessible and useful for the community. We plan to make the dataset publicly available within the next year.
>
> > While we have not yet finalized the license or specific privacy safeguards, we are actively evaluating appropriate open-access licenses and data-sharing protocols that balance openness with the protection of participant privacy. Our goal is to make a meaningful and responsible contribution to the field, and we will ensure that any released data adheres to ethical standards, including anonymization and consent requirements.
>
> **Question 2:**
> >Thank you for raising this valuable question. Testing autonomous driving models in unseen geographic domains is indeed crucial for evaluating generalization capability.
>
> >First, we would like to clarify that our model was evaluated on the Bench2Drive (NeurIPS 2024) benchmark. This benchmark not only includes CARLA Town 10, as you mentioned, but also comprises a broader range of complex and novel scenarios. Specifically, the Bench2Drive dataset covers 12 Towns from CARLA v2 (e.g., Town 05, Town 10, Town 12), using 220 routes for evaluation. Each route comes from different 12 towns and 23 weather conditions, providing a richer and more diverse set of unseen scenarios for testing. Moreover, unlike evaluations limited to driving score and success rate in a single town such as CARLA Town 10, Bench2Drive also reports performance across five advanced urban driving skills: Merging, Overtaking, Give Way, Traffic Sign, and Emergency Braking. Our supplementary materials provide the results for all of these evaluation metrics, offering a comprehensive demonstration of the generalization capabilities of **${E^3AD}$** in unseen geographic domains. You can find all the relevant details in Table 6 of the supplementary.
>
> >Second, to further validate the performance of our model in unseen geographic domains and to address your concerns, we conducted additional experiments using the nuScenes dataset. We divided the dataset by acquisition location (Singapore and Boston) into two groups: the training set consists of 12,435 and 15,695 samples for Singapore and Boston, respectively, while the test set contains 2,929 and 3,090 samples. We trained the model using data collected in one location and evaluated it on the other, ensuring that the validation involves a substantial number of truly unseen geographic domains. As shown in the table below, our model demonstrates a clear advantage over the baseline (VAD-tiny) in validation on unseen data. Especially in the case of "training on Singapore and validating on Boston", **${E^3AD}$** achieves a 14.1% reduction in baseline L2 error and a 22.7% decrease in collision rate compared to the baseline.
>
> |Scenario|L2(1s)|L2(2s)|L2(3s)|L2(Avg)|Col(1s)|Col(2s)|Col(3s)|Col(Avg)|
> |----------------------------------|------:|------:|------:|--------:|-------:|-------:|-------:|--------:|
> |Baseline(TrainingonSingapore)|0.51|0.84|1.21|0.85|0.29|0.42|0.61|0.44|
> |**${E^3AD}$**(TrainingonSingapore)|0.44|0.71|1.03|0.73|0.15|0.30|0.56|0.34|
> |Baseline(TrainingonBoston)|0.45|0.77|1.17|0.80|0.22|0.35|0.76|0.44|
> |**${E^3AD}$**(TrainingonBoston)|0.46|0.76|1.13|0.78|0.22|0.42|0.50|0.38|
>
>
> **Question 3:**
> >You have raised an interesting point regarding data scaling experiments. In fact, in our work, we have already designed specific approaches for data grouping and incremental data experiments. Specifically, we observed notable distribution differences between data from experts (who have over 10 years of driving experience and work as professional drivers) and novices (all other subjects). We visualized the EEG power spectral density (PSD) topomaps for both expert and novice groups within the frequency domain (using the Python MNE package for visualization). We then split the data into five frequency bands (i.e., delta, theta, alpha, beta, and gamma) and found that expert drivers’ power is concentrated in the occipital lobe (visual areas), whereas novice drivers’ power is concentrated in the frontal lobe (responsible for higher cognitive functions such as decision making). This is an interesting finding, suggesting that novices need to exert more cognitive effort to decide their next action, while experts rely primarily on visual processing and intuitively know what to do next.
>
> >Therefore, we grouped homogeneous data together: the expert group consists of data from 10 experts, and the novice group consists of data from 10 novices. In our scaling experiments, we incrementally expanded the data in three settings: “expert group,” “novice group,” and “expert group + novice group.” We believe this approach is more meaningful and better aligned with the focus of our study than a simple incremental scheme such as “5, 10, 15”. The results of these experiments are presented in Table 3 of our paper.
>
> ## Responses to Limitations
>
> **Limitations 1:**
> >We agree that additional generality studies would make the empirical evidence more compelling. In response, we have made the following efforts for your consideration:
> >First, we conducted experiments using GenAD as a new baseline model. It is important to note that we observed a discrepancy between the official checkpoints provided by GenAD and the performance reported in their original paper. Specifically, while the L2 error is slightly lower, the collision rate is substantially higher. We then reproduced their results using the official configuration on eight NVIDIA A100 GPUs and obtained similar findings. Therefore, our experiments reported are based on the official checkpoints. After integrating **${E^3AD}$** into GenAD, the performance of our model shows a significant improvement compared to the reproduced results of the baseline, and it also surpasses the official checkpoint. We would like to clarify that, due to the limited time available for the rebuttal, we reduced the total number of training epochs from 60 to 12. We expect that using the original number of training epochs would yield even more promising results.. These experiments further validate the generalization capability of our method.
>
> | Model| L2 (1s) | L2 (2s) | L2 (3s) | L2 (Avg) | Col (1s) | Col (2s) | Col (3s) | Col (Avg) |
> |--------------------|--------:|--------:|--------:|---------:|---------:|---------:|---------:|----------:|
> | GenAD (official)   |   0.25  |   0.46  |   0.76  |    0.49  |    0.11  |    0.21  |    0.45  |     0.26  |
> | GenAD (reproduce)  |   0.25  |   0.46  |   0.76  |    0.49  |    0.14  |    0.26  |    0.50  |     0.30  |
> | **${E^3AD}$** (GenAD)        |   0.24  |   0.44  |   0.74  |    0.47  |    0.10  |    0.21  |    0.42  |     0.24  |
>
> >Second, due to the limited rebuttal period, we commit to making further efforts to explore other recent end-to-end autonomous driving models (such as LAW) and recent benchmarks (such as NavSim), while continuing to enhance the performance of our method on GenAD. To demonstrate our ongoing commitment to improving the model, we also provide in the table below our recent efforts applying **${E^3AD}$**  to VAD. Through the expansion of EEG data and model optimization, our model’s performance has achieved a higher level. We further commit that these efforts will be fully documented in the main text or supplementary materials of the final version of the manuscript.
>
> | Model         | L2 (1s) | L2 (2s) | L2 (3s) | L2 (Avg) | Col (1s) | Col (2s) | Col (3s) | Col (Avg) |
> |---------------|--------:|--------:|--------:|---------:|---------:|---------:|---------:|----------:|
> | Baseline (VAD) |  0.41   |  0.70   |  1.05   |   0.72   |   0.07   |   0.17   |   0.41   |    0.22   |
> | **${E^3AD}$** (VAD)      |  0.40   |  0.68   |  1.03   |   0.70   |   0.04   |   0.12   |   0.30   |    0.15   |
> **Limitations 2:**
> >Please refer to the response to questions 2. We have already provided explanations and have made efforts to present more compelling results.

---

### Note · Authors · 2025-08-14

We sincerely thank all reviewers for their valuable feedback. A summary of the rebuttal is provided to assist the AC and the reviewers in their final deliberation.
## Reviewer cnpf
Unresponsive during the discussion phase.
- **Concerns**:
1. Small dataset collected and used raises concerns about over‑fitting and model bias.
2. Brain data are sensitive. A dedicated limitations subsection is needed.
3. The generalizability is yet to be explored.
4. More diverse geographic domains needs to be validated in the future.
- **Responses**:
1. Clarifying that we fine-tune our model through a contrastive learning process with LaBraM to address data scarcity.
2. Confirming full compliance with legal and ethical guidelines during data collection.
3. Supplementing experiments and results.
4. Supplementing experiments and results.

## Reviewer w1pE
All concerns have been addressed, leading to an updated score.
- **Concerns**:
1. Provide intuitive visualizations showing that the cognitive module prevents collisions.
2. Integrating E3AD into GenAD would make the empirical case far more compelling.
3. Lack of an explicit “Contributions” paragraph at the end of the Introduction.
- **Responses**:
1. Confirming results has been provided in the supplementary.
2. Supplementing experiments and results.
3. Adding the Contributions paragraph.

## Reviewer 7cU7
All concerns have been addressed, leading to an improved score.
- **Concerns**:
1. Small computational overhead lacks experimental evidence.
2. The baselines are relatively outdated.
- **Responses**:
1. Clarifying evidence presented in the Table 1.
2. Supplementing experiments and results.

## Reviewer bynr
Unresponsive during the discussion phase.
- **Concerns**:
1. Limited Alignment Validation in Stage 1.
2. Lack a mechanistic analysis of why EEG features improve planning.
3. Dataset Scalability & Release.
4. No ablation on LaBraM’s necessity.
5. Real-World Gaps.
- **Responses**:
1. Providing the loss curves and clustering analyses of model outputs across scenarios/behaviors, confirming visual–LaBraM alignment and the model’s learning of scene–behavior information.
2. Stating that the supplementary material included fine-grained metrics and analyses, while providing theoretical justification and references regarding the relationship between EEG signals and driving.
3. Committing to open-source.
4. Providing results.
5. Providing real-world data validation videos and supplementing with cross-dataset validation results.

---

### Decision · Program_Chairs · 2025-09-17

**Decision:**

Accept (poster)

**Comment:**

This paper introduces a framework which learns a “Driving-Thinking” encoder by contrastively aligning in-car video with EEG features from a pretrained brain activity model. The encoder is then frozen and integrated into existing end-to-end planners such as VAD and UniAD, without requiring EEG at inference. Results on nuScenes (open-loop) and Bench2Drive (closed-loop) show consistent improvements in L2 error, collision rate, and driving score, with negligible runtime overhead. The integration is straightforward and reproducible, and the planned release of data and code would further enhance its value.

The AC views the contribution as primarily conceptual and exploratory. The idea of incorporating human cognitive signals into end-to-end driving models is novel and interesting, but the current implementation remains early-stage. The paired EEG–video dataset is relatively small and demographically narrow, raising concerns about generalization and bias. The precise mechanisms by which EEG alignment improves planning are not deeply analyzed, and the evaluation relies heavily on open-loop nuScenes and simulation-only closed-loop tests, which do not yet capture real-world safety-critical aspects.

Overall, despite these limitations, the work is technically sound, clearly presented, and demonstrates a promising new direction for cognition-augmented learning in end-to-end driving. The AC thus recommends acceptance, and the authors are encouraged to calibrate claims, provide a transparent discussion of dataset limitations, and strengthen evaluation and analysis in future versions.